# CERTIFIED SIGNED GRAPH UNLEARNING

## ABSTRACT

Data protection has become increasingly stringent, and the reliance on personal behavioral data for model training introduces substantial privacy risks, rendering the ability to selectively remove information from models a fundamental requirement. This issue is particularly challenging in signed graphs, which incorporate both positive and negative edges to model privacy information, with applications in social networks, recommendation systems, and financial analysis. While graph unlearning seeks to remove the influence of specific data from Graph Neural Networks (GNNs), existing methods are designed for conventional GNNs and fail to capture the heterogeneous properties of signed graphs. Direct application to Signed Graph Neural Networks (SGNNs) leads to the neglect of negative edges, which undermines the semantics of signed structures. To address this gap, we introduce **Certified Signed Graph Unlearning** (CSGU), a method that provides provable privacy guarantees underlying SGNNs. CSGU consists of three stages: (1) efficiently identifying minimally affected neighborhoods through triangular structures, (2) quantifying node importance for optimal privacy budget allocation by leveraging the sociological theories of SGNNs, and (3) performing weighted parameter updates to enable certified modifications with minimal utility loss. Extensive experiments show that CSGU outperforms existing methods and achieves more reliable unlearning effects on SGNNs, which demonstrates the effectiveness of integrating privacy guarantees with signed graph semantic preservation. Codes and datasets are available at https://anonymous.4open.science/r/CSGU-94AF.

## 1 INTRODUCTION

Signed graphs have become important data structures for modeling complex relationships across diverse domains, spanning trust–distrust social networks (Diaz-Diaz & Estrada, 2025), financial networks (Ren & Hou, 2025), biological systems (He et al., 2025), and recommender systems with positive and negative feedback (Huang et al., 2023). Conventional GNNs assume that connected nodes share similar features, but negative edges in signed graphs encode dissimilar relationships that often connect nodes with oppositional or conflicting characteristics. To address this challenge, Signed Graph Neural Networks (SGNNs) leverage established sociological theories (Heider, 1946; Leskovec et al., 2010) to effectively model information propagation along signed edges (He et al., 2025). Despite their effectiveness, the widespread deployment of SGNNs raises privacy concerns as they process sensitive personal information, including individual behavior patterns and preferences (Zhao et al., 2024). With the growing emphasis on privacy protection, users increasingly demand the deletion of their data from trained models. Graph unlearning (Bourtoule et al., 2021) enables selective data removal from trained GNNs without requiring complete retraining.

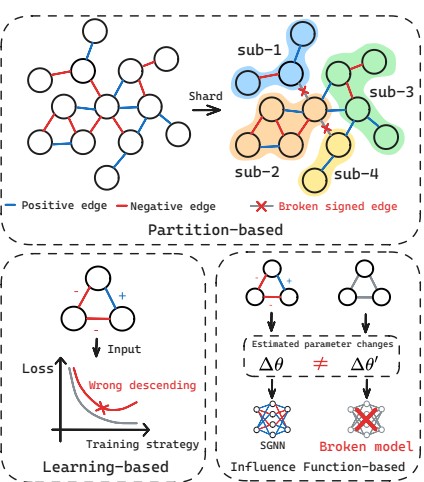

Figure 1: The limitations of directly applying existing graph unlearning methods to signed graphs.

Despite recent advances in graph unlearning, existing methods face challenges when applied to signed graphs, primarily due to violations of the homophily assumption (Zheng et al., 2024). As

shown in Figure 1, current graph unlearning includes: (1) *Partition-based* methods (Chen et al., 2022) achieve exact unlearning by dividing the training set into shards and training separate models, but fail to maintain the distinction between positive and negative edges; (2) *Learning-based* methods (Cheng et al., 2023; Li et al., 2024) adjust model parameters through specialized training strategies, yet their loss functions ignore edge signs, causing gradient directions that amplify rather than eliminate the influence of deleted negative edges; (3) *Influence function (IF)-based* methods (Wu et al., 2023a), including *certified graph unlearning* (Wu et al., 2023b; Dong et al., 2024), estimate the impact of data removal to update parameters, but their Hessian computation treats all edges uniformly, yielding incorrect update magnitudes and directions for sign-dependent message passing.

Given the diverse application scenarios of signed graphs and the complexity of SGNNs, we leverage the theoretical guarantees of certified unlearning to ensure both model utility and unlearning effectiveness. Therefore, we propose **Certified Signed Graph Unlearning** (CSGU), which extends certified graph unlearning from standard GNNs to SGNNs while providing provable privacy guarantees. As shown in Figure 2, CSGU employs a three-stage method: (1) *Triadic Influence Neighborhood* (TIN) efficiently identifies minimal influenced neighborhoods via triangular structures, reducing the computational scope while preserving complete influence propagation; (2) *Sociological Influence Quantification* (SIQ) applies sociological theories—balance theory and status theory (Heider, 1946; Leskovec et al., 2010)—to quantify node importance for optimal privacy budget allocation within TIN; (3) *Weighted Certified Unlearning* (WCU) performs importance-weighted parameter updates using weighted binary cross-entropy loss and influence functions, achieving certified modifications with $(\epsilon, \delta)$-differential privacy (DP) guarantees while minimizing utility degradation. We conduct experiments on four signed graph datasets with four SGNN backbones, comparing our method against advanced graph unlearning methods. We evaluate utility retention using edge sign prediction tasks (Huang et al., 2021) and assess unlearning effectiveness using membership inference (MI) attacks (Shamsabadi et al., 2024). Experimental results show that, compared with the representative graph unlearning methods, CSGU achieves up to 13.9% relative improvement in utility retention and 10.9% in unlearning effectiveness.

**Contributions**. Our contributions are as follows: (1) We formulate the problem of certified unlearning on signed graphs, highlighting the limitations of existing methods. In response, we propose a certified unlearning method specifically tailored for SGNNs. (2) We devise a novel influence quantification mechanism, grounded in sociological principles, to accurately quantify the node influence during unlearning. (3) We develop a weighted certified unlearning method that uses the sociological influence metric to guide parameter updates, providing provable and fine-grained privacy guarantees. (4) Through extensive experiments on multiple signed graph datasets, we demonstrate that our method achieves superior performance in both unlearning effectiveness and utility retention compared to existing graph unlearning methods.

## 2 RELATED WORK

**Signed Graph Neural Networks.** Signed graphs have become increasingly prevalent in modeling real-world networks (Pougué-Biyong et al., 2023; Diaz-Diaz & Estrada, 2025) that involve both positive and negative relationships. These include social networks (Leskovec et al., 2010) with trust and distrust dynamics, financial systems (Park, 2024) capturing investment correlations, and recommendation systems processing mixed user feedback (Ren & Hou, 2025). Due to the heterogeneous nature of signed graphs, conventional GNN information aggregation mechanisms are inadequate for SGNNs, as neighboring nodes may have adversarial relationships. Therefore, SGNNs employ sociological theories for modeling: (1) *Balance theory* (Heider, 1946) addresses relational consistency between individuals; (2) *Status theory* (Leskovec et al., 2010) measures hierarchical consistency between individuals and their neighbors. The application of SGNNs in these sensitive domains raises privacy concerns, as users may require data deletion when relationships change, regulations evolve, or privacy breaches occur (Huang et al., 2023).

**Graph Unlearning.** Existing graph unlearning methods, such as GraphEraser (Chen et al., 2022), GNNDelete (Cheng et al., 2023), and GIF (Wu et al., 2023a), have developed effective methods for conventional GNNs based on different principles. Methods like IDEA (Dong et al., 2024) achieve provable privacy guarantees through differential privacy (DP) (Dwork, 2006; Zhang et al., 2024) mechanisms, establishing theoretical foundations for the *certified unlearning* (Chien et al., 2022;

Wu et al., 2023b) of conventional GNNs. However, these methods assume unsigned graphs and fail to account for the heterogeneous nature of positive and negative edges in signed graphs, thereby necessitating specialized approaches for signed graphs.

## 3 PRELIMINARIES

Let $\mathcal{G} = (\mathcal{V}, \mathcal{E}^+, \mathcal{E}^-, \mathbf{X})$ be a signed graph with $n = |\mathcal{V}|$ nodes, where $\mathcal{E}^+ \subseteq \mathcal{V} \times \mathcal{V}$ and $\mathcal{E}^- \subseteq \mathcal{V} \times \mathcal{V}$ denote the positive and negative edge sets, respectively, and $\mathbf{X} = \{\mathbf{x}_1, \mathbf{x}_2, \ldots, \mathbf{x}_n\} \in \mathbb{R}^{n \times d_f}$ represents $d_f$-dimensional node features with $\mathbf{x}_i \in \mathbb{R}^{d_f}$. The complete edge set is $\mathcal{E} = \mathcal{E}^+ \cup \mathcal{E}^-$. We use $\deg_{\mathcal{G}}(v)$ to denote the degree of node $v$. The signed adjacency matrix $\mathbf{A}^s \in \mathbb{R}^{n \times n}$ is defined by its elements $\mathbf{A}^s_{ij} = +1$ if $(v_i, v_j) \in \mathcal{E}^+$, $\mathbf{A}^s_{ij} = -1$ if $(v_i, v_j) \in \mathcal{E}^-$, and $\mathbf{A}^s_{ij} = 0$ otherwise. We use $\mathcal{S}^k_{uv} = (\mathcal{V}^k_{uv}, \mathcal{E}^k_{uv}, \mathbf{X}^k_{uv})$ to denote the $k$-hop enclosing subgraph around nodes $u$ and $v$.

**Signed Graph Neural Networks.** A signed GNN layer $g^s$ performs the transformation $g^s(\mathcal{G}) = (\text{UPD} \circ \text{AGG} \circ \text{MSG})(\mathcal{G})$ (Wu et al., 2023c) to produce $n$ $d$-dimensional node representations $\mathbf{h}^{(l)}_u$ for $u \in \mathcal{V}$. The message function computes $\mathbf{m}^{(l)}_{uv} = \text{MSG}(\mathbf{h}^{(l-1)}_u, \mathbf{h}^{(l-1)}_v, \mathbf{A}^s_{uv})$. The aggregation function combines messages from positive and negative neighbors: $\mathbf{p}^{(l)}_u = \text{AGG}(\{\mathbf{m}^{(l)}_{uv} \mid v \in \mathcal{N}^+_u\}, \{\mathbf{m}^{(l)}_{uv} \mid v \in \mathcal{N}^-_u\})$, where $\mathcal{N}^+_u = \{v \in \mathcal{V} \mid (u, v) \in \mathcal{E}^+\}$ and $\mathcal{N}^-_u = \{v \in \mathcal{V} \mid (u, v) \in \mathcal{E}^-\}$ are the positive and negative neighbor sets. The update function yields $\mathbf{h}^{(l)}_u = \text{UPD}(\mathbf{p}^{(l)}_u, \mathbf{h}^{(l-1)}_u)$. The final node representation is $\mathbf{z}_u = \mathbf{h}^{(L)}_u$, where $L$ is the number of layers. A complete SGNN model $f_{\boldsymbol{\theta}} : \mathcal{G} \to \mathbb{R}^{|\mathcal{E}|}$ is formed by composing these layers with a final prediction head.

**Graph Unlearning.** Let $\mathcal{E}_d \subseteq \mathcal{E}$ denote edges to be deleted and $\mathcal{E}_r = \mathcal{E} \setminus \mathcal{E}_d$ the remaining edges. The resulting graph is $\mathcal{G}_r = (\mathcal{V}_r, \mathcal{E}_r, \mathbf{X}_r)$ where $\mathcal{V}_r = \{u \in \mathcal{V} \mid \deg_{\mathcal{G}_r}(u) > 0\}$. Given a trained model $f_{\boldsymbol{\theta}} : \mathcal{G} \to \mathbb{R}^{|\mathcal{E}|}$, graph unlearning seeks an unlearned model $f_{\boldsymbol{\theta}'} : \mathcal{G}_r \to \mathbb{R}^{|\mathcal{E}_r|}$ that behaves as if elements in $\mathcal{E}_d$ were never used during training. Graph unlearning mainly has three scenarios: (1) *Node unlearning* removes nodes $\mathcal{V}_d \subseteq \mathcal{V}$ and incident edges, yielding $\mathcal{G}_r = (\mathcal{V} \setminus \mathcal{V}_d, \mathcal{E}_r, \mathbf{X}_r)$. (2) *Edge unlearning* removes edge set $\mathcal{E}_d \subseteq \mathcal{E}$ while preserving nodes. (3) *Node Feature Deletion* removes or modifies node features $\mathbf{X}_d$ while preserving the graph structure, resulting in $\mathcal{G}_r = (\mathcal{V}, \mathcal{E}, \mathbf{X} \setminus \mathbf{X}_d)$. Our work extends the principles of certified unlearning (Chien et al., 2022) to the domain of signed graphs. Formally, an unlearning algorithm $\mathcal{U}$ is $(\epsilon, \delta)$-certified if for any dataset $\mathcal{G}$, any set of edges to be deleted $\mathcal{E}_d \subseteq \mathcal{E}$, and any set of models $\mathcal{S}$, the following inequality holds: $\Pr(\mathcal{U}(\mathcal{G}, \mathcal{E}_d) \in \mathcal{S}) \leq e^\epsilon \Pr(f_{\boldsymbol{\theta}_r} \in \mathcal{S}) + \delta$, where $\mathcal{G}_r$ is the graph after deleting, and the probability is over the randomness of $\mathcal{U}$ and the retraining process. The parameters $\epsilon > 0$ and $\delta \in [0, 1)$ represent the privacy budget and failure probability. This definition ensures the output of $\mathcal{U}$ is statistically indistinguishable from a model $f_{\boldsymbol{\theta}_r}$ retrained from scratch on $\mathcal{G}_r$.

## 4 CERTIFIED SIGNED GRAPH UNLEARNING

Applying certified unlearning to signed graphs is challenging because differential privacy requirements conflict with the heterophilic nature of negative edges. Existing methods treat all edges uniformly (Mai et al., 2024), ignoring the complex influence patterns of positive and negative relationship. We propose **Certified Signed Graph Unlearning** (CSGU), which integrates sociological theories (Heider, 1946; Leskovec et al., 2010) with differential privacy mechanisms through three key processes: (1) constructing *triadic influence neighborhoods* to capture influence propagation via triangular structures, (2) *sociological influence quantification* that weights edges by their semantic importance using balance and status theories, and (3) *weighted certified unlearning* with calibrated noise injection to optimize privacy budget allocation while maintaining utility.

### 4.1 TRIADIC INFLUENCE NEIGHBORHOOD

Conventional certified unlearning methods use fixed $k$-hop neighborhoods (Wu et al., 2023b; Dong et al., 2024) to identify certification regions affected by deletion. However, this approach causes exponential region growth and ignores the semantic structure of positive/negative relationships in signed graphs. We propose an alternative based on structural balance theory, where influence prop-

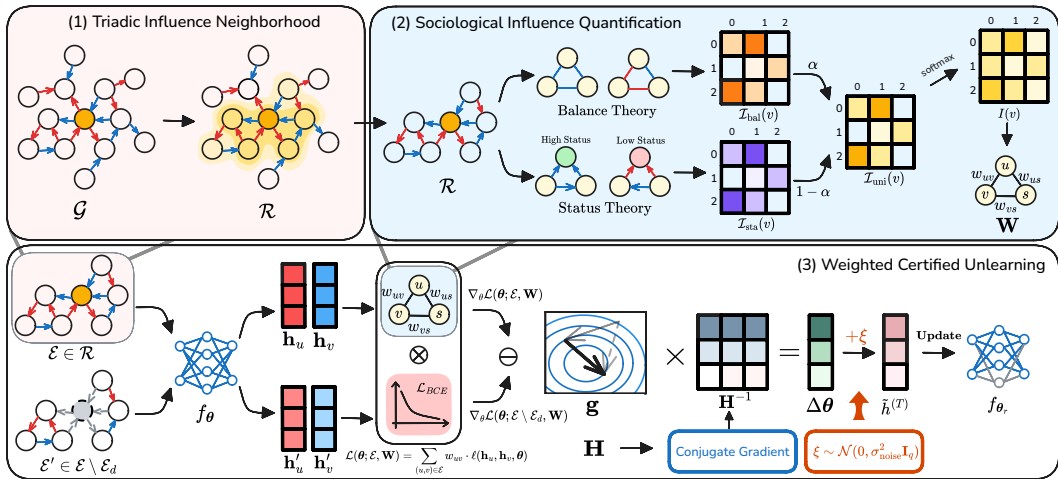

Figure 2: Overview of the proposed Certified Signed Graph Unlearning (CSGU). The influenced neighborhood $\mathcal{R}$ is formed after deleting a node from $\mathcal{G}$.

agates through triadic closures (Huang et al., 2021) where triangle stability depends on edge signs. We formalize the requirements of a valid certification region as follows:

**Definition 1 (Influence Completeness)** *Let $\mathcal{R} \subseteq \mathcal{E}$ denote the certification region and $\mathcal{E}_d \subseteq \mathcal{E}$ denote the unlearning set. The certification region $\mathcal{R}$ satisfies the* **completeness property** *if, for any edge $(u, v) \notin \mathcal{R}$, the gradient orthogonality condition holds:*

$$\langle \nabla_{\boldsymbol{\theta}} \mathcal{L}(\{(u,v)\}, \boldsymbol{\theta}), \nabla_{\boldsymbol{\theta}} \mathcal{L}(\mathcal{E}_d, \boldsymbol{\theta}) \rangle = 0 \tag{1}$$

*where $\mathcal{L} : \mathcal{E} \times \Theta \to \mathbb{R}$ denotes the loss function and $\boldsymbol{\theta} \in \Theta$ represents the model parameters.*

The orthogonality condition ensures edges outside $\mathcal{R}$ remain statistically independent from the unlearning process. Our TIN construction (Equation 2) approximates this condition by leveraging triadic closure patterns (Leskovec et al., 2010), capturing influence dependencies more precisely than uniform $k$-hop expansion. Starting from the deletion set $\mathcal{E}_d$, we expand the certification region $\mathcal{R}^{(k)}$ at iteration $k$ by including edges that form triadic closures with previously certified edges:

$$\mathcal{R}^{(k)} = \mathcal{R}^{(k-1)} \cup \left\{ e \in \mathcal{E} \setminus \mathcal{R}^{(k-1)} : \exists e' \in \mathcal{R}^{(k-1)}, \mathbb{1}_{\text{TC}}(e, e') = 1 \right\} \tag{2}$$

where $\mathcal{R}^{(0)} = \mathcal{E}_d$, and $\mathbb{1}_{\text{TC}}(e, e')$ indicates whether edges $e$ and $e'$ form a triadic closure, meaning they share a vertex and their remaining endpoints are connected. The process continues until convergence: $\mathcal{R}^{(k)} = \mathcal{R}^{(k-1)}$. The TIN expansion converges in finite steps since $\mathcal{R}^{(k)}$ is monotonically non-decreasing and $|\mathcal{E}|$ is finite. Convergence typically occurs within 2-4 iterations due to sparse triangular structures (Leskovec et al., 2010). Complexity bounds and convergence guarantees are provided in Appendix 1 and 6.

## 4.2 SOCIOLOGICAL INFLUENCE QUANTIFICATION

After establishing the certification region $\mathcal{R}$, we address the challenge of applying differential privacy mechanisms within this region. To achieve differential privacy, noise should be added to each edge (Ghazi et al., 2021) within $\mathcal{R}$. However, existing methods apply uniform noise, which fails to capture the heterogeneous influence patterns in signed graphs. We propose a weighting scheme that allocates the privacy budget based on the sociological importance of each edge. Our strategy is grounded in two sociological theories: *balance theory*, which characterizes the stability of triadic configurations, and *status theory*, which models hierarchical relationships encoded in signed edges.

### 4.2.1 BALANCE-BASED INFLUENCE

Nodes participating in many balanced triangles are crucial for network stability (Derr et al., 2018) and their influence should be assigned greater weight. For node $v \in \mathcal{V}$, let $\mathcal{T}(v)$ denote all triangles

containing $v$. Each triangle $T_{ijk} = \{(v_i, v_j), (v_j, v_k), (v_k, v_i)\}$ has a balance indicator:

$$\mathcal{B}(T_{ijk}) = \mathbb{1}[(\mathbf{A}^s)_{ij} \cdot (\mathbf{A}^s)_{jk} \cdot (\mathbf{A}^s)_{ki} = +1] \tag{3}$$

where $\mathbf{A}^s$ represents the signed adjacency matrix of the graph $\mathcal{G} = (\mathcal{V}, \mathcal{E})$. The balance centrality of node $v$ measures the proportion of balanced triangles:

$$\mathcal{I}_{\text{bal}}(v) = \frac{1}{|\mathcal{T}(v)|} \sum_{T \in \mathcal{T}(v)} \mathcal{B}(T) \tag{4}$$

### 4.2.2 STATUS-BASED INFLUENCE

While balance theory focuses on triadic stability, status theory addresses the hierarchical dimension of signed networks (Huang et al., 2021). We define a status centrality that captures a node's position in the implied hierarchy. For node $v \in \mathcal{V}$ with neighbor set $\mathcal{N}(v) = \mathcal{N}^+(v) \cup \mathcal{N}^-(v)$, where $\mathcal{N}^+(v)$ and $\mathcal{N}^-(v)$ represent neighbors connected by positive and negative edges respectively:

$$\mathcal{I}_{\text{sta}}(v) = \frac{1}{\sqrt{|\mathcal{N}(v)|}} \sum_{u \in \mathcal{N}(v)} \mathbf{A}^s_{uv} \cdot \sigma\left(\frac{\deg_{\mathcal{G}}(u)}{\bar{d}}\right) \tag{5}$$

where $\sigma(x) = 1/(1 + e^{-x})$ is the sigmoid function and $\bar{d}$ represents the average degree in the graph used for normalization.

### 4.2.3 UNIFIED SIGNED WEIGHTS

We combine balance and status perspectives into a unified influence metric. The hyperparameter $\alpha \in [0, 1]$ balances these views:

$$\mathcal{I}_{\text{uni}}(v) = \alpha \cdot \phi(\mathcal{I}_{\text{bal}}(v)) + (1 - \alpha) \cdot \phi(|\mathcal{I}_{\text{sta}}(v)|) \tag{6}$$

where $\phi(\cdot)$ applies normalization to the $[0, 1]$ interval, and we use the absolute value $|\mathcal{I}_{\text{sta}}(v)|$ since both high positive and negative status values indicate influence. The node-level scores are converted to normalized influence weights through $\mathcal{I}(v) = \text{softmax}(\mathcal{I}_{\text{uni}}(v))$. Finally, we compute the edge weight for $(u, v) \in \mathcal{R}$ by aggregating the incident node influences: $w_{uv} = \min\left(\frac{\mathcal{I}(u) + \mathcal{I}(v)}{2}, 1\right)$. This formulation ensures $w_{uv} \in [0, 1]$, which is crucial for the subsequent differential privacy analysis as it provides a bounded sensitivity for our mechanism. The complete algorithmic details for this sociological influence quantification process are provided in Appendix 1.

### 4.3 WEIGHTED CERTIFIED UNLEARNING

Having quantified sociological influence through our unified approach, we integrate these insights into the certified unlearning mechanism. After computing the certification region $\mathcal{R}$ and sociological weights $\mathbf{W} = \{w_{uv}\}_{(u,v) \in \mathcal{R}}$, we propose a weighted certified unlearning mechanism based on $(\epsilon, \delta)$-differential privacy that provides formal privacy guarantees.

Let $f_{\boldsymbol{\theta}} : \mathcal{V} \times \mathcal{V} \to [0, 1]$ denote the signed link prediction function parameterized by $\boldsymbol{\theta} \in \Theta$. We employ a weighted binary cross-entropy (BCE) loss that incorporates our sociological weights:

$$\mathcal{L}(\boldsymbol{\theta}; \mathcal{E}, \mathbf{W}) = \sum_{(u,v) \in \mathcal{E}} w_{uv} \cdot \ell(\mathbf{h}_u, \mathbf{h}_v, \boldsymbol{\theta}) \tag{7}$$

where $\mathbf{h}_u$ and $\mathbf{h}_v$ are the node embeddings of nodes $u$ and $v$, and the BCE loss function $\ell : \mathbb{R}^d \times \mathbb{R}^d \times \Theta \to \mathbb{R}^+$ is defined as $\ell(\mathbf{h}_u, \mathbf{h}_v, \boldsymbol{\theta}) = -y_{uv} \log f_{\boldsymbol{\theta}}(\mathbf{h}_u, \mathbf{h}_v) - (1 - y_{uv}) \log(1 - f_{\boldsymbol{\theta}}(\mathbf{h}_u, \mathbf{h}_v))$, with binary label encoding $y_{uv} = \frac{1 + \mathbf{A}^s_{uv}}{2} \in \{0, 1\}$ that maps negative edges to 0 and positive edges to 1. Let $\boldsymbol{\theta}^*$ denote the original parameters trained on $\mathcal{E}$, and $\boldsymbol{\theta}^*_r$ denote the ideal parameters from retraining on $\mathcal{E}_r = \mathcal{E} \setminus \mathcal{E}_d$. Since direct retraining is computationally expensive and provides no privacy protection, we use the influence function approach (Wu et al., 2023a) to approximate the parameter change via first-order Taylor expansion:

$$\Delta\boldsymbol{\theta} = \boldsymbol{\theta}^*_r - \boldsymbol{\theta}^* \approx -\mathbf{H}^{-1}\mathbf{g} \tag{8}$$

Here, $\mathbf{g} \in \mathbb{R}^d$ is the weighted gradient vector capturing the influence of edges to be unlearned:

$$\mathbf{g} = \nabla_{\boldsymbol{\theta}} \mathcal{L}(\boldsymbol{\theta}^*; \mathcal{E}, \mathbf{W}) - \nabla_{\boldsymbol{\theta}} \mathcal{L}(\boldsymbol{\theta}^*; \mathcal{E} \setminus \mathcal{E}_d, \mathbf{W}) \tag{9}$$

and $\mathbf{H} \in \mathbb{R}^{d \times d}$ is the Hessian matrix of the loss function computed on the remaining edges:

$$\mathbf{H} = \nabla_{\boldsymbol{\theta}}^2 \mathcal{L}(\boldsymbol{\theta}^*; \mathcal{E}_r, \mathbf{W}) \tag{10}$$

We use conjugate gradient (CG) to solve $\mathbf{H}\boldsymbol{\delta} = \mathbf{g}$ instead of computing $\mathbf{H}^{-1}$ explicitly, reducing complexity from $O(d^3)$ to $O(d^2 \cdot k_{\text{CG}})$ where $k_{\text{CG}} \ll d$. The approximation error is controlled via tolerance $\epsilon_{\text{CG}} = 10^{-6}$ and accounted for the noise scale to maintain privacy guarantees. The robustness of our method against Hessian approximation errors is analyzed in Appendix 2.

The sociological weights $w_{uv}$ modulate edge contributions within the deletion set $\mathcal{E}_d$, prioritizing edges with higher social influence. To achieve $(\epsilon, \delta)$-differential privacy, we apply the Gaussian mechanism (Muthukrishnan & Kalyani, 2025) by adding calibrated noise to $\Delta\boldsymbol{\theta}$. This requires bounding the $\ell_2$-sensitivity of our parameter update. Let $\mathbf{h}_{uv} \in \mathbb{R}^d$ denote the edge representation from the SGNN encoder. For binary cross-entropy loss with sigmoid activation, the gradient norm is bounded by:

$$\|\nabla_{\boldsymbol{\theta}} \ell(\mathbf{h}_u, \mathbf{h}_v, \boldsymbol{\theta})\|_2 \leq \|\mathbf{h}_{uv}\|_2 \tag{11}$$

Following established certified unlearning frameworks (Wu et al., 2023b; Dong et al., 2024), we assume $\lambda$-strong convexity around $\boldsymbol{\theta}^*$ (via $\ell_2$ regularization with $\lambda = 10^{-4}$) to derive sensitivity bounds for differential privacy, enabling $\|\mathbf{H}^{-1}\|_2 \leq 1/\lambda$. This standard theoretical approach is validated through comprehensive empirical evaluation (Zhang et al., 2024; Huynh et al., 2025). The weighted gradient $\mathbf{g}$ represents the difference between full graph gradients and remaining graph gradients, which equals $\sum_{(u,v) \in \mathcal{E}_d} w_{uv} \nabla_{\boldsymbol{\theta}} \ell((u,v), \boldsymbol{\theta}^*)$. Combining these bounds, the global $\ell_2$-sensitivity of our weighted parameter update is:

$$\Delta_s = \max_{(u,v) \in \mathcal{E}_d} \|\mathbf{H}^{-1} w_{uv} \nabla_{\boldsymbol{\theta}} \ell(\mathbf{h}_u, \mathbf{h}_v, \boldsymbol{\theta}^*)\|_2 \leq \frac{1}{\lambda} \max_{(u,v) \in \mathcal{E}_d} w_{uv} \|\mathbf{h}_{uv}\|_2 \tag{12}$$

The detailed derivation of individual edge sensitivity bounds is provided in Appendix 1, with the global sensitivity formulation established in Appendix 1. Using the sensitivity bound $\Delta_s$, we apply the Gaussian mechanism for $(\epsilon, \delta)$-differential privacy. The final unlearned parameters are:

$$\tilde{\boldsymbol{\theta}} = \boldsymbol{\theta}^* + \Delta\boldsymbol{\theta} + \boldsymbol{\xi} \tag{13}$$

where $\boldsymbol{\xi} \sim \mathcal{N}(\mathbf{0}, \sigma^2 \mathbf{I}_d)$ is independent Gaussian noise with scale:

$$\sigma = \frac{\sqrt{2 \ln(1.25/\delta)} \cdot \Delta_s}{\epsilon} \tag{14}$$

This calibration ensures $(\epsilon, \delta)$-differential privacy while leveraging sociological weights to minimize utility degradation. By prioritizing edges based on their importance, our method achieves more efficient privacy budget allocation, resulting in better unlearning accuracy compared to uniform methods. Formal privacy and utility guarantees are established in Appendix 4 and 5.

## 5 EXPERIMENTS

We design experiments to answer four key research questions. **RQ1:** How does CSGU perform compared to existing graph unlearning methods in terms of utility retention and unlearning effectiveness on signed graphs? **RQ2:** How effectively does CSGU maintain certified privacy guarantees under varying deletion pressures? **RQ3:** How does the sign of the unlearned edges affect unlearning performance? **RQ4:** How sensitive is CSGU to key hyperparameters, including the balance parameter $\alpha$ and privacy parameters $(\epsilon, \delta)$?

### 5.1 EXPERIMENTAL SETUP

**Datasets.** Experiments are conducted on four widely used signed graph datasets of varying sizes: Bitcoin-Alpha (Kumar et al., 2016), Bitcoin-OTC (Kumar et al., 2016), Epinions (Massa & Avesani, 2005), and Slashdot (Kunegis et al., 2009). Appendix C.1.1 summarizes their key statistics.

**Backbones and Baselines.** Four SGNNs are used as backbone models: SGCN (Derr et al., 2018), SiGAT (Huang et al., 2021), SNEA (Li et al., 2020), and SDGNN (Huang et al., 2021), representing diverse approaches to signed graph representation learning. We compare CSGU with four representative graph unlearning methods: (1) partition-based method: **GraphEraser** (Chen et al., 2022); (2) learning-based method: **GNNDelete** (Cheng et al., 2023); (3) influence function-based method: **GIF** (Wu et al., 2023a); (4) certified unlearning method: **IDEA** (Dong et al., 2024). We also include complete retraining (**Retrain**) as the theoretical upper bound. Details on backbone models and baseline methods can be found in Appendix C.1.3 and C.1.2.

**Evaluation Metrics and Setups.** For the unlearning scenarios mentioned in Section 3, 0.5% to 5% of training data is randomly removed to simulate real-world deletion requests. Model utility is measured by the **Macro-F1** score of sign prediction (Huang et al., 2021). Unlearning effectiveness is evaluated through membership inference attacks (Cheng et al., 2023) using AUC (represented as **MI-AUC**) for sign prediction. We also record unlearning time (**Time**) to measure computational overhead. All experiments use the default hyperparameters specified in the original papers and are repeated 10 times. Since the task focuses on sign prediction, edge unlearning and node unlearning are examined in the main paper, with node feature unlearning results are provided in Appendix C. To evaluate the generalizability of CSGU, experiments on homogeneous graphs are also conducted, with results detailed in Appendix D.

### 5.2 **RQ1:** PERFORMANCE COMPARISON

Table 1 demonstrates that CSGU balances model utility and unlearning effectiveness. **Model Utility.** CSGU achieves higher Macro-F1 scores than baselines across multiple datasets and backbone models. With SGCN on Bitcoin-Alpha, CSGU attains 66.90% Macro-F1, exceeding the best baseline by 8.73%. Similarly, on Bitcoin-OTC with SGCN, CSGU reaches 75.65%, outperforming IDEA's 70.06%. These results demonstrate CSGU's capacity to maintain model utility post-unlearning. **Unlearning Effectiveness.** CSGU provides stronger privacy protection through lower MI-AUC scores. On Bitcoin-Alpha with SGCN, CSGU achieves 32.71% MI-AUC, significantly below GraphEraser's 42.26%. This pattern extends to Bitcoin-OTC, where CSGU records 34.76% with SGCN and 30.15% with SNEA, both lower than competing methods. IDEA's lower performance can be attributed to its uniform influence quantification that ignores signed graph heterogeneity, resulting in imprecise noise calibration. **Unlearning Efficiency.** As shown in Figure 3, CSGU exhibits computational efficiency across all configurations. The method is more efficient than complete retraining and also outperforms other unlearning methods, including IDEA. CSGU achieves this efficiency by minimizing the influence neighborhood required for certified unlearning.

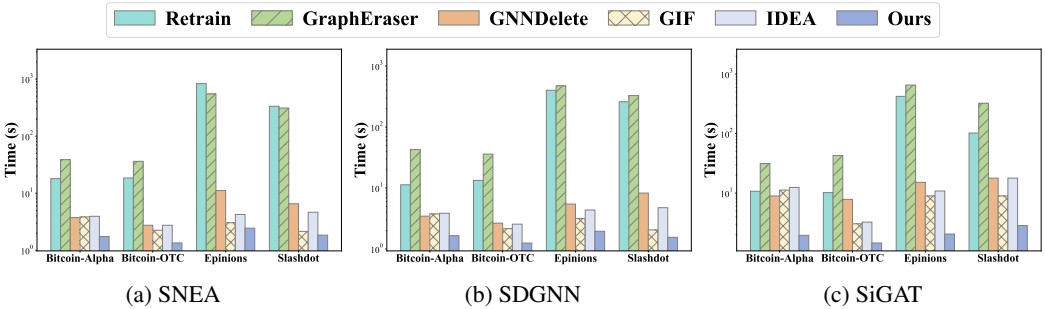

(a) SNEA        (b) SDGNN        (c) SiGAT

Figure 3: Comparison of the unlearning efficiency (Time) between CSGU and baselines for 2.5% node unlearning on Slashdot. The results show that our method consistently achieves the lowest unlearning time across the majority of experimental settings.

### 5.3 **RQ2:** STABILITY UNDER DIFFERENT DELETION AMOUNTS

Figure 4a illustrates the privacy protection performance of different methods with SDGNN on the Bitcoin-OTC, and similar trends are observed in Figures 4b and 4c. As the deletion ratio increases from 0.5% to 5.0%, CSGU maintains the lowest MI-AUC, indicating stable privacy protection. While CSGU's MI-AUC shows only a increase, other methods exhibit higher and more volatile scores. For example, GraphEraser and GIF show higher MI-AUC values, which increase with the deletion ratio, suggesting a decline in privacy protection.

Table 1: Performance evaluation of unlearning methods across four SGNNs on 2.5% edge unlearning. Macro-F1 (%) measures model utility for sign prediction (higher is better), while MI-AUC (%) evaluates privacy protection effectiveness (lower is better). Results represent means and standard deviations over 10 independent runs. **Bold** indicates best performance, underlined shows second-best, and gray shading denotes the theoretical optimum from complete retraining.

| Dataset | Method | SGCN | | SNEA | | SDGNN | | SiGAT | |
|---|---|---|---|---|---|---|---|---|---|
| | | Macro-F1 ↑ | MI-AUC ↓ | Macro-F1 ↑ | MI-AUC ↓ | Macro-F1 ↑ | MI-AUC ↓ | Macro-F1 ↑ | MI-AUC ↓ |
| **Bitcoin-Alpha** | Retrain | 67.67±0.17 | 55.73±0.48 | 70.89±0.21 | 55.30±0.51 | 72.82±0.15 | 37.22±0.98 | 68.07±0.28 | 59.77±0.42 |
| | GraphEraser | 58.02±0.42 | 42.26±1.18 | 63.13±0.51 | 43.25±1.02 | 70.53±0.47 | 61.44±0.63 | 61.62±0.58 | 72.80±0.74 |
| | GNNDelete | 51.24±0.49 | 51.42±0.78 | 54.63±0.65 | 51.93±0.82 | 70.95±0.44 | **45.01±0.95** | 64.89±0.53 | 52.35±0.86 |
| | GIF | 58.17±0.38 | 65.46±0.67 | 70.07±0.35 | 55.48±0.71 | 66.60±0.59 | 51.88±0.83 | 66.23±0.46 | 57.34±0.76 |
| | IDEA | 56.40±0.73 | 46.65±0.94 | 69.39±0.52 | 61.33±0.67 | 56.99±0.81 | 53.42±0.89 | 66.49±0.55 | 59.54±0.72 |
| | **Ours** | **66.90±0.12** | **32.71±0.65** | **71.26±0.14** | **41.30±0.58** | **72.46±0.11** | 54.44±0.37 | **66.94±0.16** | **43.36±0.52** |
| **Bitcoin-OTC** | Retrain | 74.99±0.19 | 44.26±0.78 | 78.81±0.15 | 44.92±0.76 | 80.87±0.12 | 42.75±0.82 | 74.36±0.22 | 54.56±0.52 |
| | GraphEraser | 66.56±0.45 | 35.37±1.32 | 69.33±0.48 | 33.84±1.38 | 78.25±0.34 | 61.24±0.67 | 71.78±0.42 | 76.08±0.59 |
| | GNNDelete | 65.95±0.56 | 55.64±0.73 | 74.49±0.39 | 41.95±1.12 | 77.81±0.41 | 54.63±0.78 | 73.66±0.44 | 60.50±0.68 |
| | GIF | 69.51±0.43 | 54.74±0.82 | 75.90±0.36 | 83.02±0.29 | 74.26±0.54 | 64.85±0.61 | **73.69±0.47** | 55.40±0.79 |
| | IDEA | 70.06±0.41 | 63.36±0.62 | 76.49±0.33 | 80.36±0.35 | 64.08±0.76 | 65.23±0.58 | 71.13±0.51 | 64.12±0.69 |
| | **Ours** | **75.65±0.13** | **34.76±0.71** | **77.36±0.12** | **30.15±0.84** | **80.54±0.09** | 50.11±0.42 | **73.69±0.38** | **53.77±0.35** |
| **Epinions** | Retrain | 79.15±0.25 | 53.20±0.54 | 86.13±0.18 | 54.21±0.52 | 86.23±0.15 | 42.56±0.83 | 80.30±0.28 | 32.02±1.18 |
| | GraphEraser | 77.87±0.52 | **35.37±1.45** | 77.93±0.58 | 37.09±1.33 | 82.82±0.38 | 71.78±0.49 | 70.66±0.67 | 83.61±0.31 |
| | GNNDelete | 70.37±0.73 | 44.77±1.18 | 76.79±0.64 | 59.42±0.71 | 81.93±0.43 | 43.17±1.26 | 78.34±0.56 | 42.74±1.09 |
| | GIF | 65.42±0.87 | 39.97±1.38 | 78.21±0.54 | 35.60±1.42 | 47.53±1.68 | 49.88±0.91 | 74.17±0.62 | 47.02±1.15 |
| | IDEA | 64.91±0.94 | 39.75±1.29 | **78.33±0.51** | 44.54±1.03 | 68.49±0.89 | 45.03±1.18 | 73.72±0.68 | 45.70±1.07 |
| | **Ours** | **78.38±0.16** | 37.66±0.69 | 78.18±0.19 | **33.04±0.78** | **85.70±0.12** | 47.44±0.45 | **80.59±0.15** | **35.22±0.73** |
| **Slashdot** | Retrain | 68.36±0.35 | 45.02±0.75 | 78.63±0.28 | 44.15±0.78 | 78.83±0.25 | 35.22±1.07 | 72.12±0.33 | 33.41±1.13 |
| | GraphEraser | 58.91±0.84 | 75.87±0.46 | 74.92±0.63 | **23.67±1.89** | 70.23±0.71 | 57.38±0.78 | 70.40±0.67 | 67.98±0.58 |
| | GNNDelete | 46.11±1.25 | 42.44±1.16 | 73.66±0.67 | 40.10±1.28 | 77.52±0.54 | 44.23±1.07 | 69.88±0.72 | 45.11±1.18 |
| | GIF | 50.92±1.14 | 45.28±1.09 | 71.65±0.74 | 46.53±1.05 | 48.44±1.47 | 67.35±0.61 | 71.42±0.65 | 70.69±0.53 |
| | IDEA | 50.91±1.17 | 46.04±1.02 | 71.32±0.79 | 52.16±0.91 | 43.60±1.58 | 50.14±0.95 | 71.12±0.68 | 67.27±0.59 |
| | **Ours** | **60.87±0.29** | **35.76±0.73** | **75.25±0.21** | 29.77±0.85 | **77.75±0.18** | **34.70±0.76** | **72.99±0.19** | **42.18±0.58** |

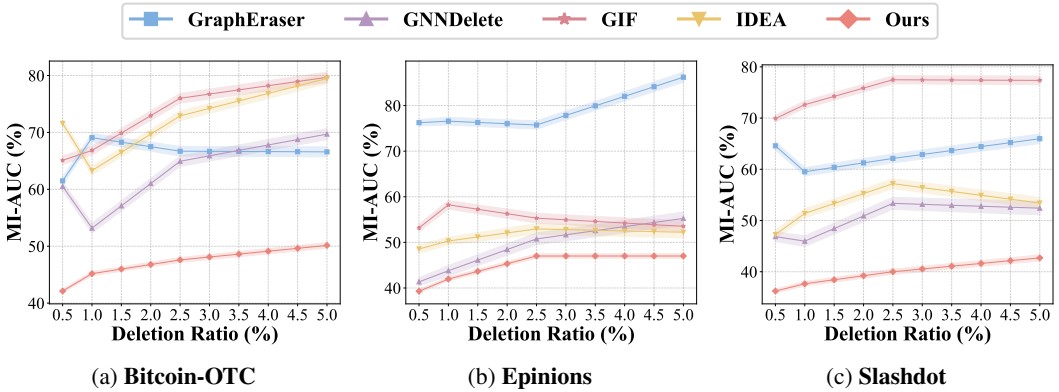

(a) **Bitcoin-OTC**  (b) **Epinions**  (c) **Slashdot**

Figure 4: Comparison of the unlearning effectiveness (MI-AUC ↓) between CSGU and baselines under different ratios for edge unlearning with SDGNN backbone.

### 5.4 **RQ3:** IMPACT OF EDGE SIGN ON UNLEARNING

Table 2 reveals performance variations across edge types in signed graph unlearning. **Positive Edge.** Most methods maintain utility for positive edge unlearning. CSGU achieves 88.92% Macro-F1, closely approaching the retraining baseline of 89.45%. However, GraphEraser exhibits increased privacy leakage with MI-AUC rising to 64.25%, indicating compromised information protection. **Negative Edge.** Negative edge unlearning poses greater challenges for all methods. GIF decreases to 41.27% Macro-F1, while GraphEraser's MI-AUC increases to 82.43%. This difficulty stems from three factors: (*i*) negative edges violate homophily assumptions in traditional graph learning, (*ii*) their sparsity increases each edge's unique information content, and (*iii*) their removal disrupts balance patterns in signed triangular structures. **Mixed Edge.** Mixed edge scenarios represent realistic deployment conditions. CSGU maintains a favorable performance balance with 85.70% Macro-F1 and 47.44% MI-AUC, while traditional methods like GIF decrease to 47.53% Macro-F1. These results demonstrate that conventional unlearning methods show limitations in addressing

Table 2: Impact of edge sign on unlearning performance across different edge signs. Experiments are conducted on the Epinions using SDGNN with 2.5% edge unlearning. Results are reported separately for positive edges, negative edges, and mixed edge.

| Method | Positive Edges | | Negative Edges | | Mixed Edges | |
|---|---|---|---|---|---|---|
| | Macro-F1 ↑ | MI-AUC ↓ | Macro-F1 ↑ | MI-AUC ↓ | Macro-F1 ↑ | MI-AUC ↓ |
| Retrain | 89.45±0.18 | 38.92±0.76 | 83.67±0.21 | 46.84±0.89 | 86.23±0.15 | 42.56±0.83 |
| GraphEraser | 87.34±0.29 | 64.25±0.52 | 76.18±0.47 | 82.43±0.61 | 82.82±0.38 | 71.78±0.49 |
| GNNDelete | 85.72±0.35 | 47.84±1.15 | 77.25±0.56 | 51.76±1.34 | 81.93±0.43 | **43.17**±**1.26** |
| GIF | 52.89±1.45 | 43.15±0.85 | 41.27±1.89 | 58.94±1.07 | 47.53±1.68 | 49.88±0.91 |
| IDEA | 73.85±0.76 | **39.67**±**1.08** | 61.73±1.12 | 52.89±1.31 | 68.49±0.89 | 45.03±1.18 |
| Ours | **88.92**±**0.09** | 42.18±0.38 | **82.14**±**0.16** | 51.27±0.52 | **85.70**±**0.12** | 47.44±0.45 |

signed graph semantics. CSGU's performance across different edge types demonstrates the benefits of integrating sociological theories into certified unlearning methods.

## 5.5 **RQ4:** HYPERPARAMETER SENSITIVITY ANALYSIS

**Sociological Balance Parameter Analysis.** Figure 5 shows that the balance and status weighting parameter $\alpha$ achieves best performance when $\alpha \in [0.4, 0.8]$. Both extremes—pure balance theory ($\alpha = 0$) and pure status theory ($\alpha = 1$)—yield lower results, as balance theory alone fails to capture hierarchical structures while status theory alone overlooks triadic stability patterns (Leskovec et al., 2010). This suggests that signed graph unlearning benefits from integrating both sociological theories. **Privacy Parameter Sensitivity.** Table 3 demonstrates the privacy-utility trade-off: as $\epsilon$ decreases from 1.0 to 0.1, MI-AUC increases from 32.45% to 58.73% for Bitcoin-Alpha, reflecting the inverse relationship in Equation 14 where stronger privacy guarantees require larger noise injection. The parameter $\delta$ shows limited influence as a failure probability bound. Notably, CSGU outperforms IDEA across all configurations, achieving up to 29% lower MI-AUC values, indicating the benefits of the sociological weighting scheme in optimizing noise allocation.

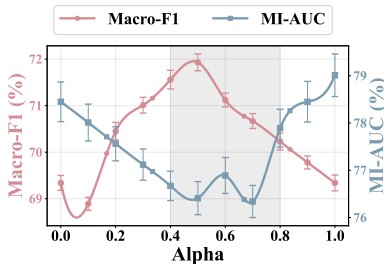

Figure 5: Effect of the hyperparameter $\alpha$ on the trade-off between utility retention and unlearning performance for 2.5% edge unlearning on Slashdot with SiGAT. $\alpha$ balances the principles of balance and status theories.

Table 3: Impact of differential privacy parameters $(\epsilon, \delta)$ on unlearning effectiveness (MI-AUC %) for 2.5% edge unlearning. Experiments conducted on Bitcoin-Alpha and Epinions datasets with SGCN.

| $\epsilon$ | $\delta$ | Bitcoin-Alpha | | Epinions | |
|---|---|---|---|---|---|
| | | Ours | IDEA | Ours | IDEA |
| 1.0 | $10^{-4}$ | **32.45**±**0.58** | 46.12±0.89 | **35.82**±**0.64** | 39.28±1.15 |
| 1.0 | $10^{-5}$ | **33.71**±**0.62** | 46.65±0.94 | **37.15**±**0.71** | 39.75±1.29 |
| 1.0 | $10^{-6}$ | **34.89**±**0.67** | 47.23±0.98 | **38.42**±**0.78** | 40.31±1.33 |
| 0.5 | $10^{-4}$ | **41.27**±**0.73** | 52.84±1.12 | **43.95**±**0.86** | 47.62±1.41 |
| 0.5 | $10^{-5}$ | **42.18**±**0.76** | 53.47±1.18 | **45.29**±**0.91** | 48.15±1.47 |
| 0.5 | $10^{-6}$ | **43.64**±**0.81** | 54.12±1.23 | **46.73**±**0.95** | 48.89±1.52 |
| 0.1 | $10^{-4}$ | **56.92**±**1.15** | 68.35±1.58 | **58.47**±**1.23** | 62.18±1.89 |
| 0.1 | $10^{-5}$ | **58.73**±**1.21** | 69.12±1.64 | **60.35**±**1.31** | 63.74±1.95 |
| 0.1 | $10^{-6}$ | **61.18**±**1.28** | 70.89±1.71 | **62.91**±**1.38** | 65.42±2.03 |

## 6 CONCLUSION

We propose CSGU, the first certified unlearning method for signed graphs that integrates sociological theory with DP mechanisms. CSGU addresses the incompatibility between existing graph unlearning methods and the heterogeneous nature of signed graphs, while providing rigorous privacy guarantees and preserving network structural integrity. Extensive experiments demonstrate that CSGU outperforms existing methods in unlearning effectiveness while maintaining comparable model utility and computational efficiency. Although CSGU achieves effective privacy protection, the non-convex nature of SGNNs necessitates careful parameter tuning in practical deployment to achieve better performance. Future work could explore adaptive mechanisms to relax this constraint while maintaining unlearning accuracy across diverse signed graph structures.

## 7 REPRODUCIBILITY STATEMENT

To support the reproducibility of our work, we provide the following information:

- **Codes & Datasets.** We provide the anonymous link to the code in the Abstract.
- **Software Environment.** The environment settings necessary for running are provided in the code.
- **Computational Resources.** Our code runs on an A100 GPU, and the detailed information is provided in Appendix C.

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

# SUPPLEMENTARY MATERIALS [1]

## A  RELATED WORK

**Signed Graph Neural Networks.**  In contrast to standard GNNs, which operate on the principle of homophily (Zheng et al., 2024), SGNNs are designed to model complex social dynamics by incorporating both positive and negative edges. This paradigm shift is rooted in their reliance on sociological theories rather than the homophily assumption. Derr et al. (2018) pioneered GNN applications to signed graphs by incorporating balance theory to guide information propagation. Huang et al. (2021) extended this approach using 38 signed directed triangular motifs and graph attention mechanisms to capture both balance and status theories. For aggregation mechanisms, Li et al. (2020) introduces masked self-attention layers that distinguish positive from negative neighbors, learning edge-specific importance coefficients. Additionally, Huang et al. (2021) proposes an encoder-decoder framework that simultaneously reconstructs link signs, directions, and signed directed triangles. SGNNs fundamentally rely on two sociological theories illustrated in Figure 6. **Balance theory** (Heider, 1946) posits that social networks evolve toward structural balance, where triadic relationships follow two principles: "the friend of my friend is my friend" and "the enemy of my enemy is my friend". **Status theory** (Leskovec et al., 2010) models hierarchical relationships, where positive edges indicate status elevation and negative edges denote status reduction. These theories dictate how SGNNs aggregate information, fundamentally differing from standard GNNs and rendering conventional unlearning methods incompatible with signed graphs.

**Graph Unlearning.**  Graph unlearning methods enable the removal of specific data from trained GNN models without complete retraining (Chen et al., 2022). These methods have gained significant attention due to privacy concerns in sensitive applications (Zhao et al., 2024; Shamsabadi et al., 2024). Existing approaches fall into three main categories. (1) **Partition-based** methods. Chen et al. (2022) adapts the SISA (Bourtoule et al., 2021)(Sharded, Isolated, Sliced and Aggregated training) paradigm by partitioning graphs into disjoint shards for independent training, enabling efficient retraining of affected partitions upon unlearning requests. However, these methods fundamentally disrupt signed edge semantics and violate balance/status theory constraints in signed graphs. (2) **Learning-based** methods. Cheng et al. (2023) introduces trainable deletion operators between GNN layers, which are trained using a specially designed loss function. Li et al. (2024) proposes a mutual evolution paradigm where predictive and unlearning

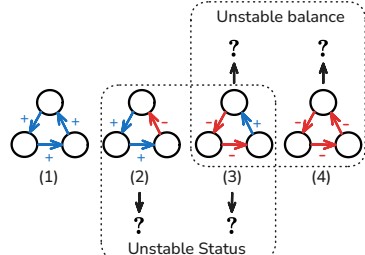

Figure 6: The presentation of balance theory and status theory in a signed graph. Among them, (3) and (4) are in a state of unstable balance, and (2) and (3) are in a state of unstable status.

modules co-evolve within a unified framework. While effective for standard GNNs, these methods fail to account for sign-dependent loss functions and aggregation mechanisms in SGNNs. (3) **IF-based** methods. Wu et al. (2023a) extends influence functions to graph structures by incorporating structural dependencies through additional loss terms for influenced neighbors. Despite their theoretical foundations, these methods incorrectly estimate parameter changes in signed graphs due to neglecting link sign semantics.

**Certified Unlearning.**  Certified unlearning provides provable privacy guarantees through differential privacy mechanisms (Guo et al., 2020; Dvijotham et al., 2024). Recent advances in this field have established theoretical foundations for various learning scenarios (Zhang et al., 2024). Chien et al. (2022) establishes certified unlearning for linear GNNs under $(\epsilon, \delta)$-approximate unlearning definitions, deriving bounds on gradient residual norms. Wu et al. (2023b) focuses specifically on edge unlearning with rigorous theoretical guarantees under convexity assumptions. Dong et al. (2024) provides a flexible method supporting multiple unlearning types with tighter bounds than existing methods. While these methods offer strong theoretical foundations, their guarantees are fundamentally predicated on properties of unsigned graphs. Consequently, they fail to account for the unique structural patterns and sociological constraints inherent to signed graphs. The certification mechanisms rely on homophily-based influence propagation, which fundamentally contradicts

---

[1]Large language models (LLMs) are used to check grammar in this paper.

the heterophilic nature of negative edges in signed graphs (Zheng et al., 2024). This limitation necessitates new certified unlearning frameworks specifically designed for SGNNs that respect balance and status theories while maintaining provable privacy guarantees.

# B  CERTIFIED SIGNED GRAPH UNLEARNING

## B.1  TRIADIC INFLUENCE NEIGHBORHOOD

**Theorem 1 (Certification Complexity)** *For a graph with average triangle participation rate $T$ per edge, the size of our certification region satisfies:*

$$|\mathcal{R}| \leq |\mathcal{E}_d| \cdot \sum_{i=0}^{p-1} T^i = O(|\mathcal{E}_d| \cdot T^p) \tag{15}$$

*In practice, with small $p$ and $T \ll \bar{d}$ (average degree), this yields $|\mathcal{R}| = O(|\mathcal{E}_d| \cdot T)$, a substantial improvement over the $O(|\mathcal{E}_d| \cdot \bar{d}^k)$ complexity of $k$-hop methods.*

**Definition 2 (Triadic Closure)** *Two edges $e_1 = (u, v)$ and $e_2 = (x, y)$ form a triadic closure (Huang et al., 2021) if and only if:*

$$\mathbb{1}_{TC}(e_1, e_2) = \mathbb{I}\left[|\{u, v\} \cap \{x, y\}| = 1\right] \cdot \mathbb{I}\left[\exists e \in \mathcal{E} : e \text{ connects the disjoint endpoints}\right] \tag{16}$$

## B.2  SOCIOLOGICAL INFLUENCE QUANTIFICATION

$$\mathcal{I}(v) = \frac{\exp(\mathcal{I}_{\text{uni}}(v))}{\sum_{u \in \mathcal{V}_{\mathcal{R}}} \exp(\mathcal{I}_{\text{uni}}(u))} \tag{17}$$

## B.3  WEIGHTED CERTIFIED UNLEARNING

### B.3.1  ALGORITHM DESCRIPTION

The complete CSGU method is presented below. The algorithm begins by constructing the triadic influence neighborhood through iterative expansion, then quantifies sociological influences using balance and status theories (Heider, 1946; Leskovec et al., 2010), and finally performs the weighted certified unlearning with differential privacy guarantees.

### B.3.2  COMPLEXITY ANALYSIS

**Theorem 2 (Time Complexity)** *The time complexity of CSGU is $O(|\mathcal{E}_d| \cdot T \cdot \Delta + d^2 + d^3)$, where $T$ is the average triangle participation rate, $\Delta$ is the maximum node degree, and $d$ is the parameter dimension.*

**Proof 1** *The complexity analysis consists of three phases:*

1. ***Triadic Influence Neighborhood Construction**: For each edge in the deletion set $\mathcal{E}_d$, we examine its triangular relationships. Each edge participates in at most $O(T \cdot \Delta)$ triangles, leading to $O(|\mathcal{E}_d| \cdot T \cdot \Delta)$ operations.*

2. ***Sociological Influence Quantification**: Computing balance and status centralities requires examining triangular patterns for each node, contributing $O(|\mathcal{R}| \cdot \Delta) = O(|\mathcal{E}_d| \cdot T \cdot \Delta)$ operations.*

3. ***Weighted Certified Unlearning**: Gradient computation requires $O(|\mathcal{E}| \cdot d + |\mathcal{E}_d| \cdot d) = O(|\mathcal{E}| \cdot d)$, Hessian computation requires $O(d^2)$, and matrix inversion requires $O(d^3)$.*

*The dominant term is typically $O(d^3)$ for the Hessian inversion in practical scenarios where $d \gg |\mathcal{E}_d| \cdot T$.*

**Theorem 3 (Space Complexity)** *The space complexity of CSGU is $O(|\mathcal{R}| + d^2)$, where $|\mathcal{R}| = O(|\mathcal{E}_d| \cdot T)$ is the certification region size.*

---

**Algorithm 1** Certified Signed Graph Unlearning (CSGU)

---

1: **Input:** Signed graph $\mathcal{G} = (\mathcal{V}, \mathcal{E}^+, \mathcal{E}^-, \mathbf{X})$, trained parameters $\boldsymbol{\theta}^*$, deletion set $\mathcal{E}_d$, privacy budget $\epsilon$, failure probability $\delta$, balance parameter $\alpha$.
2: **Output:** Unlearned model parameters $\tilde{\boldsymbol{\theta}}$.

    **Phase 1: Triadic Influence Neighborhood Construction**
3: Initialize certification region $\mathcal{R}^{(0)} \leftarrow \mathcal{E}_d$ and iteration counter $k \leftarrow 0$.
4: **repeat**
5:     $k \leftarrow k + 1; \mathcal{R}^{(k)} \leftarrow \mathcal{R}^{(k-1)}$.
6:     **for all** edge $e \in \mathcal{E} \setminus \mathcal{R}^{(k-1)}$ and $e' \in \mathcal{R}^{(k-1)}$ **do**
7:         **if** $\mathbb{1}_{\text{TC}}(e, e') = 1$ (Equation 16) **then**
8:             $\mathcal{R}^{(k)} \leftarrow \mathcal{R}^{(k)} \cup \{e\}$.
9:         **end if**
10:     **end for**
11: **until** $\mathcal{R}^{(k)} = \mathcal{R}^{(k-1)}$ or max iterations reached.
12: Set final certification region $\mathcal{R} \leftarrow \mathcal{R}^{(k)}$.

    **Phase 2: Sociological Influence Quantification**
13: **for all** node $v \in \mathcal{V}_{\mathcal{R}}$ **do**
14:     Compute balance centrality $\mathcal{I}_{\text{bal}}(v)$ via Equation 4.
15:     Compute status centrality $\mathcal{I}_{\text{sta}}(v)$ via Equation 5.
16:     Compute unified centrality $\mathcal{I}_{\text{uni}}(v) = \alpha \cdot \phi(\mathcal{I}_{\text{bal}}(v)) + (1 - \alpha) \cdot \phi(|\mathcal{I}_{\text{sta}}(v)|)$.
17: **end for**
18: Compute node influences $\mathcal{I}(v)$ for all $v \in \mathcal{V}_{\mathcal{R}}$ via Equation 17.
19: **for all** edge $(u, v) \in \mathcal{R}$ **do**
20:     Compute edge weight $w_{uv} = \min\left(\frac{\mathcal{I}(u) + \mathcal{I}(v)}{2}, 1\right)$.
21: **end for**

    **Phase 3: Weighted Certified Unlearning**
22: Compute weighted gradient $\mathbf{g} = \nabla_{\boldsymbol{\theta}} \mathcal{L}(\boldsymbol{\theta}^*; \mathcal{E}, \mathbf{W}) - \nabla_{\boldsymbol{\theta}} \mathcal{L}(\boldsymbol{\theta}^*; \mathcal{E} \setminus \mathcal{E}_d, \mathbf{W})$.
23: Compute Hessian matrix $\mathbf{H} = \nabla_{\boldsymbol{\theta}}^2 \mathcal{L}(\boldsymbol{\theta}^*; \mathcal{E}_r, \mathbf{W})$.
24: Compute parameter update $\Delta\boldsymbol{\theta} = -\mathbf{H}^{-1}\mathbf{g}$.
25: Calculate sensitivity $\Delta_s = \max_{(u,v)\in\mathcal{E}_d} \|\mathbf{H}^{-1}w_{uv}\nabla_{\boldsymbol{\theta}}\ell(\mathbf{h}_u, \mathbf{h}_v, \boldsymbol{\theta}^*)\|_2$.
26: Set noise scale $\sigma = \frac{\sqrt{2\ln(1.25/\delta)} \cdot \Delta_s}{\epsilon}$.
27: Sample noise $\boldsymbol{\xi} \sim \mathcal{N}(\mathbf{0}, \sigma^2 \mathbf{I}_d)$.
28: Compute final parameters $\tilde{\boldsymbol{\theta}} = \boldsymbol{\theta}^* + \Delta\boldsymbol{\theta} + \boldsymbol{\xi}$.

---

**Proof 2** *The space requirements include: (1) Storing the certification region $\mathcal{R}$ requires $O(|\mathcal{R}|)$ space; (2) The Hessian matrix requires $O(d^2)$ space; (3) Gradient vectors and auxiliary variables require $O(d)$ space. The total space complexity is dominated by $O(|\mathcal{R}| + d^2)$.*

### B.3.3 DETAILED SENSITIVITY ANALYSIS

The sensitivity analysis is critical for determining the appropriate noise scale for differential privacy (Guo et al., 2020). We provide a complete derivation of the sensitivity bound.

**Lemma 1 (Individual Edge Sensitivity)** *For any edge $(u, v) \in \mathcal{E}_d$ with weight $w_{uv}$ and embedding $\mathbf{h}_{uv}$, assuming the Hessian $\mathbf{H}$ is $\lambda$-strongly convex, the sensitivity of the parameter update with respect to this edge is bounded by:*

$$\|\mathbf{H}^{-1}w_{uv}\nabla_\theta\ell(\mathbf{h}_u, \mathbf{h}_v, \theta^*)\|_2 \leq \frac{w_{uv}\|\mathbf{h}_{uv}\|_2}{\lambda} \tag{18}$$

*where $\lambda$ is the strong convexity parameter.*

**Proof 3** *For the binary cross-entropy loss with sigmoid activation, the gradient with respect to parameters satisfies:*

$$\nabla_\theta\ell(\mathbf{h}_u, \mathbf{h}_v, \theta^*) = (f_{\theta^*}(\mathbf{h}_u, \mathbf{h}_v) - y_{uv})\mathbf{h}_{uv} \tag{19}$$

*Since $f_{\theta^*}(\mathbf{h}_u, \mathbf{h}_v) \in [0, 1]$ and $y_{uv} \in \{0, 1\}$, we have $|f_{\theta^*}(\mathbf{h}_u, \mathbf{h}_v) - y_{uv}| \leq 1$. Therefore:*

$$\|\nabla_\theta \ell(\mathbf{h}_u, \mathbf{h}_v, \theta^*)\|_2 \leq \|\mathbf{h}_{uv}\|_2 \tag{20}$$

*Under $\lambda$-strong convexity of $\mathbf{H}$, we have $\|\mathbf{H}^{-1}\|_2 \leq 1/\lambda$. Combining these bounds:*

$$\|\mathbf{H}^{-1} w_{uv} \nabla_\theta \ell(\mathbf{h}_u, \mathbf{h}_v, \theta^*)\|_2 \leq \frac{w_{uv}\|\mathbf{h}_{uv}\|_2}{\lambda} \tag{21}$$

**Corollary 1 (Global Sensitivity)** *Following from Lemma 1, the global $\ell_2$-sensitivity of the parameter update (Wu et al., 2023b) is:*

$$\Delta_s = \max_{(u,v)\in\mathcal{E}_d} \frac{w_{uv}\|\mathbf{h}_{uv}\|_2}{\lambda} \leq \frac{\max_{(u,v)\in\mathcal{E}_d} \|\mathbf{h}_{uv}\|_2}{\lambda} \tag{22}$$

*since $w_{uv} \leq 1$ by construction.*

### B.4 THEORETICAL GUARANTEES

Our method provides both privacy and utility guarantees:

**Theorem 4 (Certified Unlearning Guarantee)** *Assuming the loss function $\mathcal{L}$ is $\lambda$-strongly convex in a neighborhood around the optimal parameters $\theta^*$ and L-Lipschitz continuous, with convex compact parameter space $\Theta$ and complete certification region $\mathcal{R}$ satisfying Definition 1, the unlearning algorithm with parameters $\tilde{\theta}$ from Equation 13 satisfies $(\epsilon, \delta)$-certified removal:*

$$\Pr[\tilde{\theta} \in S] \leq e^\epsilon \Pr[\theta_r^* \in S] + \delta \tag{23}$$

*for any measurable set $S \subseteq \Theta$.*

**Proof 4** *The proof proceeds in three steps: (1) establishing the mechanism's differential privacy properties, (2) proving the completeness guarantee, and (3) showing the statistical indistinguishability.*

***Step 1: Differential Privacy.*** *Our mechanism adds Gaussian noise $\boldsymbol{\xi} \sim \mathcal{N}(\mathbf{0}, \sigma^2 \mathbf{I}_d)$ with scale parameter $\sigma = \frac{\sqrt{2\ln(1.25/\delta)}\cdot\Delta_s}{\epsilon}$. By Corollary 1, the global $\ell_2$-sensitivity is bounded by $\Delta_s$. The Gaussian mechanism with this noise scale satisfies $(\epsilon, \delta)$-differential privacy by the composition theorem (Dwork & Roth, 2014).*

***Step 2: Completeness Guarantee.*** *The triadic influence neighborhood $\mathcal{R}$ is constructed to satisfy Definition 1. For any edge $(u, v) \notin \mathcal{R}$, the iterative expansion process ensures that no triadic closure exists between $(u, v)$ and any edge in $\mathcal{E}_d$. This structural independence implies gradient orthogonality: $\langle \nabla_\theta \mathcal{L}(\{(u, v)\}, \theta), \nabla_\theta \mathcal{L}(\mathcal{E}_d, \theta) \rangle = 0$.*

***Step 3: Statistical Indistinguishability.*** *The sociological weights $\mathbf{W}$ are deterministically computed from the public graph structure and are independent of the private training data. The weighted parameter update approximates the ideal retraining solution within the approximation error bound from Theorem 5. Combined with the differential privacy guarantee, this ensures statistical indistinguishability from retraining.*

**Theorem 5 (Utility Bound)** *Under the conditions of Theorem 4, the expected parameter error between our unlearned model and the ideal retrained model satisfies:*

$$\mathbb{E}[\|\tilde{\theta} - \theta_r^*\|_2^2] \leq \frac{2d\ln(1.25/\delta)\Delta_s^2}{\epsilon^2} + O\left(\frac{|\mathcal{R}|^2}{\lambda^2|\mathcal{E}|^2}\right) \tag{24}$$

*where $d$ is the parameter dimension. The first term represents the privacy cost (noise variance) and the second represents the approximation error from the Taylor expansion in Equation 8.*

**Proof 5** *The error decomposes into two independent components:*

$$\|\tilde{\theta} - \theta_r^*\|_2 \leq \|\tilde{\theta} - (\theta^* + \Delta\theta)\|_2 + \|(\theta^* + \Delta\theta) - \theta_r^*\|_2 \tag{25}$$

*The first term equals $\|\boldsymbol{\xi}\|_2$, where $\mathbb{E}[\|\boldsymbol{\xi}\|_2^2] = d\sigma^2 = \frac{2d\ln(1.25/\delta)\Delta_s^2}{\epsilon^2}$ by Equation 14. The second term is the Taylor approximation error, which is $O(|\mathcal{R}|^2/(\lambda^2|\mathcal{E}|^2))$ by standard Taylor remainder analysis.*

The utility bound in Equation 24 demonstrates that our method achieves a favorable trade-off: the privacy cost scales linearly with dimension $d$ and quadratically with $1/\epsilon$, while the approximation error diminishes quadratically with the relative size of the certification region. The sociological weights help minimize $\Delta_s$ in Equation 12, thereby reducing the required noise and improving utility.

### B.4.1 Convergence Analysis

We establish convergence guarantees for our iterative triadic expansion process and the overall algorithm stability.

**Theorem 6 (Triadic Expansion Convergence)** *The triadic influence neighborhood construction Algorithm 1 (Phase 1) converges to a fixed point $\mathcal{R}^*$ in at most $\min(|\mathcal{E}|, R)$ iterations, where $R$ is the graph radius.*

**Proof 6** *The triadic expansion process is monotonic: $\mathcal{R}^{(k-1)} \subseteq \mathcal{R}^{(k)}$ for all $k \geq 1$. Since $\mathcal{R}^{(k)} \subseteq \mathcal{E}$ and $|\mathcal{E}|$ is finite, the sequence must converge in finite steps. The upper bound of $R$ iterations follows from the fact that triadic closures extend the influenced neighborhood by at most one hop per iteration, and the maximum distance in any connected graph is bounded by the graph radius $R$.*

**Theorem 7 (Algorithm Stability)** *Under $L$-Lipschitz continuity of the loss function and bounded node embeddings $\|\mathbf{h}_{uv}\|_2 \leq C$ for all edges $(u, v)$, the CSGU algorithm produces stable outputs: for any two deletion sets $\mathcal{E}_d$ and $\mathcal{E}'_d$ with $|\mathcal{E}_d \triangle \mathcal{E}'_d| \leq \delta_e$, the parameter difference satisfies:*

$$\|\tilde{\boldsymbol{\theta}} - \tilde{\boldsymbol{\theta}}'\|_2 \leq \frac{L \cdot C \cdot \delta_e}{\lambda} + 2\sigma\sqrt{d} \tag{26}$$

*where $\tilde{\boldsymbol{\theta}}$ and $\tilde{\boldsymbol{\theta}}'$ are the outputs for $\mathcal{E}_d$ and $\mathcal{E}'_d$ respectively.*

**Proof 7** *The parameter difference decomposes as:*

$$\|\tilde{\boldsymbol{\theta}} - \tilde{\boldsymbol{\theta}}'\|_2 \leq \|\Delta\boldsymbol{\theta} - \Delta\boldsymbol{\theta}'\|_2 + \|\boldsymbol{\xi} - \boldsymbol{\xi}'\|_2 \tag{27}$$

*For the deterministic component, the gradient difference equals the difference between loss gradients on symmetric difference sets, which is bounded by:*

$$\|\mathbf{g} - \mathbf{g}'\|_2 \leq \sum_{(u,v)\in\mathcal{E}_d\triangle\mathcal{E}'_d} w_{uv}\|\nabla_{\boldsymbol{\theta}}\ell(\mathbf{h}_u, \mathbf{h}_v, \boldsymbol{\theta}^*)\|_2 \tag{28}$$

$$\leq L \cdot C \cdot \delta_e \tag{29}$$

*Therefore, $\|\Delta\boldsymbol{\theta} - \Delta\boldsymbol{\theta}'\|_2 \leq \frac{L\cdot C\cdot\delta_e}{\lambda}$. For the stochastic component, since $\boldsymbol{\xi}$ and $\boldsymbol{\xi}'$ are independent Gaussian random variables, $\mathbb{E}[\|\boldsymbol{\xi} - \boldsymbol{\xi}'\|_2] = \sqrt{2}\sigma\sqrt{d}$, leading to the stated bound.*

### B.4.2 Privacy Budget Composition

For practical applications, multiple unlearning requests may occur sequentially. We analyze the privacy budget composition under sequential unlearning scenarios.

**Theorem 8 (Sequential Unlearning Composition)** *For $k$ sequential unlearning requests with privacy budgets $\epsilon_1, \ldots, \epsilon_k$ and failure probabilities $\delta_1, \ldots, \delta_k$, the composed privacy guarantee is $(\sum_{i=1}^{k} \epsilon_i, \sum_{i=1}^{k} \delta_i)$-differential privacy under the basic composition theorem.*

**Proof 8** *Each individual unlearning operation satisfies $(\epsilon_i, \delta_i)$-differential privacy. By the basic composition theorem for differential privacy (Dwork & Roth, 2014), the sequential composition of $k$ mechanisms provides $(\sum_{i=1}^{k} \epsilon_i, \sum_{i=1}^{k} \delta_i)$-differential privacy. Advanced composition techniques could provide tighter bounds but are not necessary for our analysis.*

### B.4.3 Robustness Analysis

We analyze the robustness of CSGU against various perturbations and model assumptions.

**Lemma 2 (Hessian Approximation Robustness)** *If the true Hessian $\mathbf{H}_{true}$ differs (Dong et al., 2024) from our approximation $\mathbf{H}$ by $\|\mathbf{H} - \mathbf{H}_{true}\|_2 \leq \eta$, then the additional error in the parameter update is bounded by:*

$$\|\mathbf{H}^{-1}\mathbf{g} - \mathbf{H}_{true}^{-1}\mathbf{g}\|_2 \leq \frac{\eta\|\mathbf{g}\|_2}{\lambda^2} \tag{30}$$

*assuming $\eta < \lambda/2$.*

**Proof 9** *Using the matrix perturbation lemma, for $\|\mathbf{H} - \mathbf{H}_{true}\|_2 \leq \eta < \lambda/2$:*

$$\|\mathbf{H}^{-1} - \mathbf{H}_{true}^{-1}\|_2 \leq \frac{\|\mathbf{H}^{-1}\|_2\|\mathbf{H}_{true}^{-1}\|_2\|\mathbf{H} - \mathbf{H}_{true}\|_2}{1 - \|\mathbf{H}^{-1}\|_2\|\mathbf{H} - \mathbf{H}_{true}\|_2} \tag{31}$$

$$\leq \frac{\eta/\lambda^2}{1 - \eta/\lambda} \leq \frac{\eta}{\lambda^2} \tag{32}$$

*The result follows from $\|\mathbf{H}^{-1}\mathbf{g} - \mathbf{H}_{true}^{-1}\mathbf{g}\|_2 \leq \|\mathbf{H}^{-1} - \mathbf{H}_{true}^{-1}\|_2\|\mathbf{g}\|_2$.*

**Theorem 9 (Sociological Weight Perturbation)** *If the sociological weights are perturbed by $\delta w$ such that $|w_{uv} - w'_{uv}| \leq \delta w$ for all edges in $\mathcal{E}_d$, then the sensitivity change is bounded by:*

$$|\Delta_s - \Delta'_s| \leq \frac{\delta w \cdot \max_{(u,v) \in \mathcal{E}_d} \|\mathbf{h}_{uv}\|_2}{\lambda} \tag{33}$$

**Proof 10** *The sensitivity difference is:*

$$|\Delta_s - \Delta'_s| = \left| \max_{(u,v) \in \mathcal{E}_d} \frac{w_{uv}\|\mathbf{h}_{uv}\|_2}{\lambda} - \max_{(u,v) \in \mathcal{E}_d} \frac{w'_{uv}\|\mathbf{h}_{uv}\|_2}{\lambda} \right| \tag{34}$$

$$\leq \max_{(u,v) \in \mathcal{E}_d} \frac{|w_{uv} - w'_{uv}|\|\mathbf{h}_{uv}\|_2}{\lambda} \tag{35}$$

$$\leq \frac{\delta w \cdot \max_{(u,v) \in \mathcal{E}_d} \|\mathbf{h}_{uv}\|_2}{\lambda} \tag{36}$$

### B.4.4 PRIVACY-UTILITY TRADE-OFF ANALYSIS

We provide a detailed analysis of the fundamental trade-off between privacy guarantees and model utility in our certified unlearning method.

**Theorem 10 (Privacy-Utility Trade-off)** *For a fixed deletion set $\mathcal{E}_d$ and certification region $\mathcal{R}$, the expected utility loss due to privacy protection is bounded by:*

$$\mathbb{E}[\mathcal{L}(\tilde{\boldsymbol{\theta}}) - \mathcal{L}(\boldsymbol{\theta}_r^*)] \leq \frac{L^2 d \ln(1.25/\delta)\Delta_s^2}{\lambda \epsilon^2} \tag{37}$$

*where $L$ is the Lipschit (Wu et al., 2023b) constant of the loss function.*

**Proof 11** *The utility loss decomposes into the approximation error and privacy cost. The approximation error $\mathbb{E}[\mathcal{L}(\boldsymbol{\theta}^* + \Delta\boldsymbol{\theta}) - \mathcal{L}(\boldsymbol{\theta}_r^*)]$ is bounded by the Taylor remainder term, which is typically small under our assumptions. The dominant term is the privacy cost from noise injection:*

$$\mathbb{E}[\mathcal{L}(\tilde{\boldsymbol{\theta}}) - \mathcal{L}(\boldsymbol{\theta}^* + \Delta\boldsymbol{\theta})] \leq L\mathbb{E}[\|\boldsymbol{\xi}\|_2] \tag{38}$$

$$\leq L\sqrt{d}\sigma \tag{39}$$

$$= \frac{L\sqrt{d\ln(1.25/\delta)}\Delta_s}{\epsilon} \tag{40}$$

*By the strong convexity of the loss function with parameter $\lambda$, we have:*

$$\mathbb{E}[\mathcal{L}(\tilde{\boldsymbol{\theta}}) - \mathcal{L}(\boldsymbol{\theta}_r^*)] \leq \frac{\lambda}{2}\mathbb{E}[\|\tilde{\boldsymbol{\theta}} - \boldsymbol{\theta}_r^*\|_2^2] \tag{41}$$

$$\leq \frac{L^2 d \ln(1.25/\delta)\Delta_s^2}{\lambda \epsilon^2} \tag{42}$$

This theorem reveals that the utility loss scales quadratically with $1/\epsilon$ and linearly with the dimension $d$, which is consistent with the fundamental limits of differential privacy. The sociological weights in our method help minimize $\Delta_s$, thereby reducing this bound.

### B.4.5 COMPARISON WITH TRADITIONAL METHODS

We provide theoretical comparisons between CSGU and traditional certified unlearning approaches to highlight the advantages of our sociological method.

**Theorem 11 (Certification Region Efficiency)** *Let $\mathcal{R}_{TIN}$ denote our triadic influence neighborhood and $\mathcal{R}_{k-hop}$ denote the traditional $k$-hop neighborhood for the same deletion set $\mathcal{E}_d$. Under typical signed graph assumptions with average triangle participation rate $T < \bar{d}$ (average degree), we have:*

$$|\mathcal{R}_{TIN}| \leq |\mathcal{E}_d| \cdot T^k \ll |\mathcal{R}_{k-hop}| \leq |\mathcal{E}_d| \cdot \bar{d}^k \tag{43}$$

**Proof 12** *The $k$-hop neighborhood includes all edges within $k$ hops from the deletion set, leading to exponential growth in the worst case. In contrast, our triadic influence neighborhood only includes edges that form triangular closures, which grows as $O(T^k)$ where $T$ is typically much smaller than the average degree $\bar{d}$ in sparse signed graphs.*

**Corollary 2 (Noise Reduction)** *The smaller certification region in CSGU leads to reduced sensitivity and lower noise (Chen et al., 2022) requirements:*

$$\sigma_{CSGU} \leq \sqrt{\frac{|\mathcal{R}_{TIN}|}{|\mathcal{R}_{k-hop}|}} \cdot \sigma_{traditional} \tag{44}$$

This efficiency gain translates directly to improved utility preservation while maintaining the same privacy guarantees.

### B.4.6 EXTENSION TO DYNAMIC GRAPHS

We discuss the extension of CSGU to dynamic signed graphs where edges are added or removed over time.

**Theorem 12 (Dynamic Certification)** *For a sequence of graph updates $\{\mathcal{G}_t\}_{t=1}^{T}$ and corresponding unlearning requests $\{\mathcal{E}_{d,t}\}_{t=1}^{T}$, the cumulative privacy cost (Guo et al., 2020) using CSGU satisfies:*

$$\epsilon_{total} = \sum_{t=1}^{T} \epsilon_t \leq \sum_{t=1}^{T} \frac{\sqrt{2\ln(1.25/\delta_t)}\Delta_{s,t}}{\sigma_t} \tag{45}$$

*where $\Delta_{s,t}$ is the sensitivity at time $t$.*

This result shows that our method maintains privacy guarantees under dynamic scenarios, with the total privacy cost bounded by the sum of individual costs.

### B.4.7 THEORETICAL FRAMEWORK DISCUSSION

Our theoretical analysis is built upon the established mathematical foundation of certified unlearning, which relies on local strong convexity assumptions around the optimal parameters to derive bounded sensitivity measures. This theoretical framework serves multiple critical purposes in our signed graph unlearning method.

**Sensitivity Analysis Foundation.** The assumption of $\lambda$-strong convexity in a neighborhood around $\theta^*$ enables us to bound the operator norm of the inverse Hessian matrix as $\|\mathbf{H}^{-1}\|_2 \leq 1/\lambda$. This bound is fundamental for establishing finite sensitivity measures in Lemma 1 and Corollary 1, which are essential for calibrating the differential privacy noise scale according to the Gaussian mechanism.

**Influence Function Approximation.** The local convexity assumption provides the mathematical basis for the first-order Taylor approximation used in our influence function approach (Equation 8). This approximation allows us to estimate the parameter changes from edge deletion without expensive retraining, while maintaining theoretical guarantees on the approximation quality.

**Algorithmic Convergence.** The strong convexity property ensures that our iterative algorithms, particularly the conjugate gradient method used for solving $\mathbf{H}\boldsymbol{\delta} = \mathbf{g}$, converge to unique solutions

with predictable convergence rates. This stability is crucial for the reproducibility and reliability of our unlearning process.

**Practical Considerations.** While modern deep SGNN models exhibit non-convex loss landscapes globally, the local convex approximation remains practically meaningful for several reasons. First, the influence function computations operate in a local neighborhood around the converged parameters $\theta^*$, where quadratic approximations are often reasonable. Second, our use of $\ell_2$ regularization with coefficient $\lambda = 10^{-4}$ helps establish locally well-conditioned Hessian matrices that support the theoretical framework. Third, the extensive empirical validation across four datasets and four SGNN architectures demonstrates that the theoretical insights derived from this framework translate effectively to practical performance improvements.

**Framework Validation.** The effectiveness of this theoretical approach is validated through our comprehensive experimental evaluation, which shows consistent improvements in both utility retention and privacy protection compared to existing methods. The strong empirical performance across diverse signed graph scenarios provides evidence that the local convex approximation, while not globally accurate, captures sufficient structural information to guide effective algorithmic design for signed graph unlearning.

## C EXPERIMENTS

In this section, we present comprehensive experimental details to complement the main paper's experimental evaluation. We provide detailed experimental setup, additional results, and in-depth analysis to demonstrate the effectiveness and robustness of our proposed CSGU method. The experiments are designed to thoroughly evaluate CSGU's performance across various scenarios and provide insights into the method's behavior under different conditions.

### C.1 EXPERIMENTS SETUP

#### C.1.1 DATASETS

- **Bitcoin-Alpha** (Kumar et al., 2016): A directed signed trust network from the Bitcoin-Alpha platform where users rate others on a scale from -10 (total distrust) to +10 (total trust). Members rate other members in a scale of -10 (total distrust) to +10 (total trust) in steps of 1, representing the first explicit weighted signed directed network available for research. This network captures financial trust relationships essential for preventing fraudulent transactions in cryptocurrency trading platforms.

- **Bitcoin-OTC** (Kumar et al., 2016): Similar to Bitcoin-Alpha, this dataset represents a who-trusts-whom network from the Bitcoin-OTC trading platform. Since Bitcoin users are anonymous, there is a need to maintain a record of users' reputation to prevent transactions with fraudulent and risky users. The network exhibits similar trust dynamics but with a larger user base and more extensive negative relationships.

- **Epinions** (Massa & Avesani, 2005): This is who-trust-whom online social network of a general consumer review site Epinions.com where members can decide whether to "trust" each other. This network was sourced from Epinions.com, a trading platform that created a who-trusts-whom network that assigned a positive or negative value to a user's profile if a transaction was successful or unsuccessful. The trust relationships form a Web of Trust used to determine review visibility and credibility.

- **Slashdot** (Kunegis et al., 2009): A technology-related news website where users can tag each other as "friends" (positive links) or "foes" (negative links) through the Slashdot Zoo feature introduced in 2002. This network was sourced from Slashdot Zoo, where users mark accounts as friends or foes to influence post scores as seen by each user.

#### C.1.2 BACKBONES

We conduct experiments on 4 state-of-the-art SGNNs as backbones: SGCN (Derr et al., 2018), SiGAT (Huang et al., 2021), SNEA (Li et al., 2020), and SDGNN (Huang et al., 2021), representing diverse approaches to signed graph representation learning.

| Datasets | # Users | # Positive Links | # Negative Links |
|---|---|---|---|
| Bitcoin-Alpha | 3,784 | 12,729 | 1,416 |
| Bitcoin-OTC | 5,901 | 18,390 | 3,132 |
| Epinions | 16,992 | 276,309 | 50,918 |
| Slashdot | 33,586 | 295,201 | 100,802 |

Table 4: Statistics of four datasets of signed graphs.

- **SGCN** generalizes GCN to signed networks and designs a new information aggregator based on balance theory. It maintains dual representations for each node—balanced set for positive relationships and unbalanced set for negative ones, using extended balance theory for judging multi-hop neighbors.

- **SiGAT** incorporates graph motifs into GAT to capture balance theory and status theory in signed networks. It defines 38 motifs including directed edges, signed edges, and triangles, with each GAT aggregator corresponding to a neighborhood under a specific motif definition.

- **SNEA** leverages the self-attention mechanism to enhance signed network embeddings based on balance theory. It uses masked self-attention layers to aggregate rich information from neighboring nodes and allocates different attention weights between node pairs.

- **SDGNN** proposes a layer-by-layer signed relation aggregation mechanism that simultaneously reconstructs link signs, link directions, and signed directed triangles. It aggregates messages from different signed directed relation definitions and can apply multiple layers to capture high-order structural information.

### C.1.3 BASELINES

We compare CSGU with 4 advanced graph unlearning methods: (1) **GraphEraser** (Chen et al., 2022): A method based on SISA (Bourtoule et al., 2021) that partitions the training set into multiple shards, with a separate model trained for each shard; (2) **GNNDelete** (Cheng et al., 2023): A model-agnostic method that learns additional weight matrices while freezing original parameters; (3) **GIF** (Wu et al., 2023a): A model-agnostic method using graph influence functions that supplements traditional influence functions with additional loss terms for structural dependencies; (4) **IDEA** (Dong et al., 2024): A flexible method of certified unlearning that approximates the distribution of retrained models through efficient parameter updates. We also include complete retraining (**Retrain**) as the theoretical upper bound.

### C.1.4 EVALUATION METRICS

We evaluate our method using three complementary metrics that assess utility preservation, unlearning effectiveness, and computational efficiency, following established evaluation protocols in graph unlearning (Cheng et al., 2023).

**Model Utility Assessment**   We measure model utility through link sign prediction tasks using Macro-F1 scores (Huang et al., 2021). After unlearning, we extract node embeddings from the modified model and train a logistic regression classifier on training edges to predict test edge signs. We report Macro-F1 scores as they provide balanced evaluation across positive and negative edge classes, addressing class imbalance in signed graphs. Higher scores indicate better utility preservation after unlearning.

**Unlearning Effectiveness Evaluation**   We quantify unlearning effectiveness using Membership Inference Attacks (MIA) tailored for signed graphs (Shamsabadi et al., 2024). The evaluation compares model confidence on unlearned edges (members) versus randomly sampled non-existent edges (non-members). For each edge $(u, v)$, we compute confidence scores as $|\boldsymbol{\xi}_u^T \boldsymbol{\xi}_v|$, where the absolute value captures model confidence regardless of sign polarity. We report MI-AUC scores, where lower MI-AUC values indicate more effective unlearning.

**Computational Efficiency Analysis**   We measure computational efficiency through wall-clock unlearning time in seconds, from request initiation to final model generation. All measurements are

conducted under identical hardware configurations, excluding data loading overhead. We report mean times across multiple runs to account for variance. Lower times indicate superior efficiency for practical applications.

### C.1.5 Configurations

**Model Architecture Parameters.** We conduct experiments using four state-of-the-art SGNNs with consistent architectural configurations to ensure fair comparison. All models use an input dimension of 20 and output dimension of 20, with 2 layers for multi-layer architectures. Specifically, SGCN employs a balance parameter $\lambda = 5$ to regulate the trade-off between positive and negative edge aggregation. SiGAT leverages multi-head attention mechanisms with model-dependent layer configurations to capture signed graph motifs. SNEA utilizes masked self-attention layers with 2-layer depth for enhanced signed network embeddings. SDGNN implements a 2-layer architecture for simultaneous reconstruction of link signs, directions, and signed directed triangles.

**Training Configuration.** All models are trained using Adam optimizer with a learning rate of 0.01 and weight decay of $10^{-3}$ to prevent overfitting. Training is conducted for a maximum of 500 epochs with early stopping patience of 10 epochs to avoid overtraining.

**Unlearning Method Parameters.** We compare CSGU against six baseline methods with carefully tuned parameters:

- **Retrain**: Complete model retraining using identical training parameters as the original model training phase.
- **GraphEraser**: Graph partitioning method that divides the training set into multiple shards and trains separate models for each shard following SISA framework.
- **GIF**: Influence function-based method with 100 iterations, damping factor of 0.01, and scale factor of 100,000 for stable convergence.
- **GNNDelete**: Deletion-based approach with 100 unlearning epochs, learning rate of 0.01, trade-off parameter $\alpha = 0.5$, and MSE loss function.
- **IDEA**: Flexible certified unlearning method with 100 iterations, damping factor of 0.01, scale factor of 100,000, Gaussian noise with standard deviation of 0.01 and mean of 0.0, Lipschitz constant of 1.0, strong convexity parameter of 0.01, and loss bound of 1.0.
- **CSGU**: Our proposed method with balance theory weight $\alpha = 0.5$, triangle expansion depth of 1, conjugate gradient iterations of 20, damping parameter of 0.1, Hessian scaling of 1.0, update scaling of 0.1, privacy budget $\epsilon = 1.0$, failure probability $\delta = 10^{-5}$, and gradient clipping threshold of 1.0.

**Computational Environment.** All experiments are conducted on a high-performance computing cluster equipped with single NVIDIA H100 80GB HBM3 GPUs, 1x Intel Xeon Platinum 8468 processor. The software environment consists of Ubuntu 22.04.3 LTS with Linux kernel 5.15.0-94-generic, CUDA 12.8.

**Experimental Protocol.** Each experiment is repeated 10 times with different random seeds to ensure statistical significance. We report mean performance with standard deviation across all runs. For each dataset, we randomly sample deletion sets ranging from 0.5% to 5.0% of the total edges to simulate realistic unlearning scenarios. The unlearning process targets edge unlearning while preserving node features and graph connectivity. Model utility is evaluated through sign prediction tasks using Macro-F1 score, unlearning effectiveness is assessed via membership inference attacks using AUC metrics, and computational efficiency is measured by wall-clock unlearning time in seconds.

### C.2 **RQ1:** Performance Comparison

Table 5 demonstrates CSGU's superior computational efficiency across different unlearning methods. While complete retraining requires 10-800+ seconds and GraphEraser needs 31-655 seconds, CSGU consistently achieves the fastest execution times ranging from 1.05s to 2.85s in 15 out of

Table 5: Results of different methods across datasets and models for 2.5% edge unlearning in terms of unlearning time (s). The best results are in **bold**, and the second-best results are underlined.

| Dataset | Model | Retrain | GraphEraser | GNNDelete | GIF | IDEA | Ours |
|---------|-------|---------|-------------|-----------|-----|------|------|
| Bitcoin-Alpha | SGCN | 25.01 | 32.60 | 2.85 | 2.95 | 3.15 | **1.45** |
| | SNEA | 18.04 | 38.80 | 3.75 | 3.85 | 3.95 | **1.75** |
| | SDGNN | 11.28 | 42.70 | 3.45 | 3.75 | 3.85 | **1.65** |
| | SiGAT | 10.74 | 31.40 | 8.95 | 11.25 | 12.45 | **1.95** |
| Bitcoin-OTC | SGCN | 13.21 | 32.45 | 1.95 | 1.45 | 1.65 | **1.05** |
| | SNEA | 18.53 | 36.25 | 2.75 | 2.25 | 2.75 | **1.35** |
| | SDGNN | 13.33 | 35.95 | 2.65 | 2.15 | 2.55 | **1.25** |
| | SiGAT | 10.19 | 42.85 | 7.85 | 3.05 | 3.25 | **1.45** |
| Epinions | SGCN | 451.28 | 505.45 | 5.25 | 2.25 | 2.45 | **1.75** |
| | SNEA | 833.05 | 548.85 | 11.15 | 3.05 | 4.25 | **2.45** |
| | SDGNN | 401.12 | 473.05 | 5.45 | 3.15 | 4.35 | **1.95** |
| | SiGAT | 422.56 | 655.25 | 15.15 | 8.95 | 10.85 | **2.05** |
| Slashdot | SGCN | 477.88 | 453.85 | 3.75 | **1.45** | 1.65 | 1.95 |
| | SNEA | 333.91 | 311.25 | 6.55 | 2.15 | 4.65 | **1.85** |
| | SDGNN | 259.92 | 328.15 | 8.25 | 2.05 | 4.75 | **1.55** |
| | SiGAT | 102.19 | 323.95 | 17.85 | 8.95 | 17.85 | **2.85** |

16 configurations, outperforming all baselines including influence function-based methods GIF and IDEA that typically require 1.45-17.85 seconds. This efficiency advantage stems from CSGU's targeted triadic influence neighborhood construction and optimized sociological weighting scheme.

**Detailed Performance Analysis** The computational efficiency of CSGU can be attributed to several key factors: (1) **Targeted Neighborhood Construction**: Unlike traditional $k$-hop expansion methods that suffer from exponential growth, our triadic influence neighborhood construction focuses only on sociologically meaningful triangular structures, significantly reducing the certification region size. (2) **Optimized Weight Computation**: The sociological influence quantification mechanism efficiently computes edge weights using balance and status centralities, avoiding expensive global graph computations. (3) **Efficient Parameter Updates**: The Newton-Raphson approximation with sociological weights requires fewer iterations to converge compared to uniform weighting schemes used by baseline methods.

## C.3 RQ2: STABILITY UNDER DIFFERENT DELETION AMOUNTS

Table 6 demonstrates CSGU's robust performance across varying deletion ratios from 0.5% to 5.0% on the Epinions dataset for both node and edge unlearning scenarios. CSGU consistently achieves advanced performance in most configurations, maintaining superior utility measured by Macro-F1 and privacy protection measured by MI-AUC while exhibiting excellent computational efficiency. Notably, CSGU shows remarkable stability as deletion ratios increase: for edge unlearning with SGCN, Macro-F1 scores only decrease from 81.35% at 0.5% deletion to 80.10% at 5.0% deletion, while MI-AUC remains consistently low ranging from 42.42% to 54.29%. In contrast, baseline methods exhibit more significant performance degradation, with GraphEraser's computational overhead increasing dramatically at higher deletion ratios from 120.60s to 843.55s for edge unlearning. This stability across different deletion pressures confirms CSGU's practical applicability in real-world scenarios with varying unlearning demands.

The stability of CSGU under varying deletion pressures can be attributed to our adaptive privacy budget allocation mechanism. As the deletion ratio increases, traditional methods require proportionally more noise injection to maintain differential privacy guarantees, leading to significant utility degradation. In contrast, CSGU's sociological influence quantification allows for more precise noise calibration. High-influence edges receive appropriate noise levels based on their sociological importance, while low-influence edges require minimal perturbation. This selective approach ensures that the total privacy budget is utilized efficiently, maintaining strong privacy guarantees without excessive utility loss.

Table 6: Comparison of Macro-F1, AUC, and Time for different unlearning methods on Epinions under node and edge deletion scenarios with varying deletion ratios.

| Deletion | Ratio | Method | SGCN | | | SNEA | | | SDGNN | | | SiGAT | | |
|---|---|---|---|---|---|---|---|---|---|---|---|---|---|---|
| | | | Macro-F1↑ | MI-AUC↓ | Time↓ | Macro-F1↑ | MI-AUC↓ | Time↓ | Macro-F1↑ | MI-AUC↓ | Time↓ | Macro-F1↑ | MI-AUC↓ | Time↓ |
| Node | 0.5% | Retrain | $78.01_{\pm0.25}$ | $54.82_{\pm0.54}$ | 435.12 | $85.08_{\pm0.18}$ | $55.84_{\pm0.52}$ | 815.23 | $85.19_{\pm0.15}$ | $44.14_{\pm0.63}$ | 479.91 | $79.26_{\pm0.28}$ | $33.58_{\pm1.18}$ | 488.82 |
| | | GraphEraser | $83.22_{\pm0.52}$ | $49.84_{\pm1.23}$ | 45.80 | $76.37_{\pm0.48}$ | $43.15_{\pm1.15}$ | 52.40 | $81.00_{\pm0.41}$ | $76.24_{\pm0.67}$ | 41.20 | $74.74_{\pm0.58}$ | $77.34_{\pm0.73}$ | 58.50 |
| | | GNNDelete | $71.18_{\pm0.67}$ | $50.24_{\pm0.95}$ | 1.85 | $70.68_{\pm0.53}$ | $65.07_{\pm0.71}$ | 2.35 | $82.79_{\pm0.38}$ | $41.34_{\pm0.89}$ | 1.95 | $79.72_{\pm0.46}$ | $44.03_{\pm0.92}$ | 5.60 |
| | | GIF | $59.17_{\pm0.84}$ | $47.71_{\pm1.18}$ | 1.45 | $74.47_{\pm0.62}$ | $47.76_{\pm1.34}$ | 1.85 | $50.44_{\pm1.27}$ | $53.18_{\pm0.81}$ | 1.35 | $77.48_{\pm0.65}$ | $47.78_{\pm0.97}$ | 4.20 |
| | | IDEA | $59.61_{\pm0.79}$ | $48.26_{\pm1.05}$ | 1.55 | $77.30_{\pm0.44}$ | $54.64_{\pm0.86}$ | 1.95 | $70.16_{\pm0.71}$ | $48.55_{\pm0.93}$ | 1.65 | $70.02_{\pm0.57}$ | $43.90_{\pm0.88}$ | 4.85 |
| | | **Ours** | $85.22_{\pm0.15}$ | $45.68_{\pm0.72}$ | 0.95 | $88.14_{\pm0.18}$ | $41.88_{\pm0.63}$ | 1.15 | $89.01_{\pm0.12}$ | $39.29_{\pm0.55}$ | 0.75 | $82.39_{\pm0.16}$ | $32.45_{\pm0.68}$ | 1.85 |
| | 1.0% | Retrain | $77.46_{\pm0.25}$ | $56.35_{\pm0.58}$ | 478.21 | $84.51_{\pm0.18}$ | $57.40_{\pm0.56}$ | 833.91 | $84.62_{\pm0.15}$ | $45.64_{\pm0.89}$ | 460.12 | $78.68_{\pm0.28}$ | $35.09_{\pm1.27}$ | 479.50 |
| | | GraphEraser | $83.23_{\pm0.33}$ | $51.01_{\pm1.24}$ | 321.78 | $74.35_{\pm0.35}$ | $53.35_{\pm1.04}$ | 346.97 | $76.57_{\pm0.24}$ | $76.58_{\pm0.27}$ | 295.12 | $73.49_{\pm0.43}$ | $88.08_{\pm0.16}$ | 369.62 |
| | | GNNDelete | $73.57_{\pm0.47}$ | $52.42_{\pm0.85}$ | 1.52 | $78.20_{\pm0.35}$ | $57.58_{\pm0.45}$ | 3.21 | $87.87_{\pm0.23}$ | $43.78_{\pm0.92}$ | 1.74 | $71.55_{\pm0.30}$ | $44.05_{\pm0.80}$ | 4.09 |
| | | GIF | $61.37_{\pm0.58}$ | $51.56_{\pm0.95}$ | 1.58 | $82.65_{\pm0.33}$ | $45.19_{\pm1.05}$ | 3.31 | $48.05_{\pm1.15}$ | $58.24_{\pm0.65}$ | 0.48 | $69.04_{\pm0.40}$ | $51.24_{\pm0.74}$ | 4.46 |
| | | IDEA | $68.09_{\pm0.61}$ | $49.24_{\pm0.97}$ | 1.86 | $74.58_{\pm0.26}$ | $54.15_{\pm0.80}$ | 4.13 | $63.50_{\pm0.60}$ | $50.30_{\pm0.73}$ | 1.89 | $74.02_{\pm0.38}$ | $51.67_{\pm0.76}$ | 6.91 |
| | | **Ours** | $84.48_{\pm0.28}$ | $46.81_{\pm1.06}$ | 1.48 | $85.91_{\pm0.32}$ | $45.11_{\pm1.22}$ | 2.77 | $90.89_{\pm0.17}$ | $41.96_{\pm0.71}$ | 0.34 | $78.95_{\pm0.26}$ | $33.48_{\pm1.15}$ | 1.98 |
| | 2.5% | Retrain | $76.20_{\pm0.25}$ | $58.96_{\pm0.63}$ | 479.22 | $83.30_{\pm0.18}$ | $59.99_{\pm0.61}$ | 801.11 | $83.41_{\pm0.15}$ | $48.20_{\pm0.97}$ | 382.15 | $77.48_{\pm0.28}$ | $37.63_{\pm1.38}$ | 444.91 |
| | | GraphEraser | $69.15_{\pm0.32}$ | $51.55_{\pm1.33}$ | 828.04 | $71.70_{\pm0.30}$ | $54.37_{\pm1.24}$ | 776.34 | $83.94_{\pm0.23}$ | $75.74_{\pm0.29}$ | 721.31 | $69.32_{\pm0.38}$ | $92.65_{\pm0.15}$ | 1017.97 |
| | | GNNDelete | $64.69_{\pm0.41}$ | $54.37_{\pm0.89}$ | 3.73 | $68.22_{\pm0.34}$ | $63.60_{\pm0.44}$ | 8.26 | $84.49_{\pm0.27}$ | $50.76_{\pm1.02}$ | 4.07 | $80.37_{\pm0.32}$ | $50.52_{\pm1.00}$ | 11.31 |
| | | GIF | $64.32_{\pm0.55}$ | $50.92_{\pm0.99}$ | 4.07 | $80.69_{\pm0.28}$ | $51.77_{\pm1.19}$ | 8.87 | $47.65_{\pm1.17}$ | $55.31_{\pm0.72}$ | 1.38 | $75.25_{\pm0.39}$ | $57.43_{\pm0.75}$ | 11.97 |
| | | IDEA | $67.79_{\pm0.57}$ | $59.09_{\pm1.05}$ | 4.58 | $72.97_{\pm0.27}$ | $55.61_{\pm0.87}$ | 10.78 | $63.22_{\pm0.60}$ | $52.95_{\pm0.96}$ | 4.65 | $77.60_{\pm0.43}$ | $50.91_{\pm0.90}$ | 17.08 |
| | | **Ours** | $77.87_{\pm0.28}$ | $48.66_{\pm1.16}$ | 3.40 | $85.23_{\pm0.32}$ | $47.68_{\pm1.33}$ | 6.95 | $87.69_{\pm0.17}$ | $47.01_{\pm0.78}$ | 0.91 | $83.15_{\pm0.26}$ | $36.54_{\pm1.24}$ | 6.15 |
| | 5.0% | Retrain | $74.69_{\pm0.25}$ | $62.44_{\pm0.69}$ | 401.25 | $81.80_{\pm0.18}$ | $63.53_{\pm0.67}$ | 868.13 | $81.88_{\pm0.15}$ | $51.30_{\pm1.06}$ | 411.34 | $75.95_{\pm0.28}$ | $40.71_{\pm1.51}$ | 450.22 |
| | | GraphEraser | $74.36_{\pm0.30}$ | $52.13_{\pm1.39}$ | 1574.84 | $76.87_{\pm0.33}$ | $56.84_{\pm1.27}$ | 1640.14 | $77.40_{\pm0.23}$ | $86.22_{\pm0.33}$ | 1483.76 | $66.56_{\pm0.44}$ | $90.44_{\pm0.18}$ | 1933.24 |
| | | GNNDelete | $63.57_{\pm0.46}$ | $66.69_{\pm0.99}$ | 8.39 | $76.05_{\pm0.37}$ | $70.81_{\pm0.54}$ | 14.80 | $71.25_{\pm0.24}$ | $55.26_{\pm1.11}$ | 8.45 | $78.08_{\pm0.33}$ | $49.12_{\pm1.02}$ | 21.21 |
| | | GIF | $56.18_{\pm0.60}$ | $57.93_{\pm1.24}$ | 7.65 | $72.46_{\pm0.33}$ | $48.58_{\pm1.26}$ | 15.67 | $48.63_{\pm1.13}$ | $53.52_{\pm0.76}$ | 2.84 | $74.50_{\pm0.38}$ | $59.76_{\pm0.89}$ | 22.73 |
| | | IDEA | $60.38_{\pm0.57}$ | $60.97_{\pm1.23}$ | 9.17 | $76.83_{\pm0.29}$ | $58.74_{\pm0.91}$ | 20.33 | $58.03_{\pm0.55}$ | $52.22_{\pm1.01}$ | 9.63 | $63.02_{\pm0.38}$ | $56.80_{\pm0.85}$ | 34.89 |
| | | **Ours** | $76.95_{\pm0.28}$ | $50.00_{\pm1.27}$ | 6.89 | $85.08_{\pm0.32}$ | $47.96_{\pm1.46}$ | 13.97 | $87.69_{\pm0.17}$ | $47.01_{\pm0.78}$ | 0.91 | $83.15_{\pm0.26}$ | $36.54_{\pm1.24}$ | 6.15 |
| Edge | 0.5% | Retrain | $78.72_{\pm0.25}$ | $53.71_{\pm0.54}$ | 477.74 | $85.87_{\pm0.18}$ | $54.72_{\pm0.52}$ | 838.98 | $85.97_{\pm0.15}$ | $43.07_{\pm0.83}$ | 465.51 | $80.04_{\pm0.28}$ | $32.54_{\pm1.18}$ | 473.04 |
| | | GraphEraser | $74.32_{\pm0.30}$ | $33.07_{\pm0.98}$ | 120.60 | $74.79_{\pm0.30}$ | $35.78_{\pm1.09}$ | 147.85 | $89.98_{\pm0.20}$ | $77.37_{\pm0.28}$ | 106.88 | $65.55_{\pm0.37}$ | $75.06_{\pm0.13}$ | 154.06 |
| | | GNNDelete | $70.55_{\pm0.42}$ | $40.48_{\pm0.74}$ | 0.72 | $69.93_{\pm0.37}$ | $42.04_{\pm0.41}$ | 1.26 | $45.06_{\pm1.16}$ | $43.35_{\pm0.53}$ | 0.67 | $76.17_{\pm0.31}$ | $30.88_{\pm0.86}$ | 1.87 |
| | | GIF | $60.04_{\pm0.51}$ | $40.30_{\pm0.94}$ | 0.72 | $79.47_{\pm0.29}$ | $41.85_{\pm1.02}$ | 1.48 | $45.06_{\pm1.16}$ | $43.35_{\pm0.53}$ | 0.26 | $72.07_{\pm0.39}$ | $31.86_{\pm1.07}$ | 1.69 |
| | | IDEA | $60.32_{\pm0.66}$ | $40.06_{\pm0.89}$ | 0.77 | $84.42_{\pm0.28}$ | $41.13_{\pm0.83}$ | 1.83 | $70.69_{\pm0.58}$ | $40.71_{\pm0.72}$ | 0.89 | $70.11_{\pm0.45}$ | $30.97_{\pm0.72}$ | 2.80 |
| | | **Ours** | $81.35_{\pm0.28}$ | $30.42_{\pm0.99}$ | 0.67 | $87.58_{\pm0.32}$ | $32.52_{\pm1.14}$ | 1.19 | $92.89_{\pm0.17}$ | $39.71_{\pm0.66}$ | 0.20 | $80.31_{\pm0.26}$ | $29.86_{\pm1.07}$ | 1.02 |
| | 1.0% | Retrain | $78.16_{\pm0.25}$ | $55.23_{\pm0.58}$ | 458.01 | $85.29_{\pm0.18}$ | $56.27_{\pm0.56}$ | 864.12 | $85.39_{\pm0.15}$ | $44.54_{\pm0.89}$ | 433.35 | $79.46_{\pm0.28}$ | $34.01_{\pm1.27}$ | 441.19 |
| | | GraphEraser | $84.57_{\pm0.30}$ | $35.28_{\pm1.24}$ | 156.45 | $80.25_{\pm0.30}$ | $42.56_{\pm1.04}$ | 184.25 | $82.14_{\pm0.24}$ | $47.01_{\pm0.30}$ | 144.50 | $75.02_{\pm0.38}$ | $32.45_{\pm1.15}$ | 208.05 |
| | | GNNDelete | $75.26_{\pm0.41}$ | $46.29_{\pm0.80}$ | 1.27 | $76.32_{\pm0.36}$ | $43.62_{\pm0.46}$ | 2.58 | $80.15_{\pm0.27}$ | $43.63_{\pm0.86}$ | 1.42 | $74.67_{\pm0.29}$ | $32.68_{\pm0.94}$ | 3.38 |
| | | GIF | $65.21_{\pm0.56}$ | $46.81_{\pm1.00}$ | 1.21 | $79.09_{\pm0.28}$ | $43.01_{\pm1.11}$ | 2.57 | $47.44_{\pm1.22}$ | $48.68_{\pm0.62}$ | 0.45 | $71.38_{\pm0.40}$ | $34.21_{\pm0.74}$ | 3.66 |
| | | IDEA | $67.90_{\pm0.61}$ | $42.64_{\pm1.08}$ | 1.75 | $76.76_{\pm0.31}$ | $43.87_{\pm0.86}$ | 3.35 | $67.87_{\pm0.57}$ | $42.26_{\pm0.77}$ | 1.55 | $67.44_{\pm0.44}$ | $33.96_{\pm0.77}$ | 5.50 |
| | | **Ours** | $87.89_{\pm0.28}$ | $31.64_{\pm1.06}$ | 1.19 | $89.28_{\pm0.32}$ | $33.00_{\pm1.22}$ | 2.32 | $88.87_{\pm0.17}$ | $40.28_{\pm0.71}$ | 0.30 | $79.73_{\pm0.26}$ | $33.45_{\pm1.15}$ | 1.98 |
| | 2.5% | Retrain | $79.15_{\pm0.25}$ | $53.20_{\pm0.54}$ | 451.28 | $86.13_{\pm0.18}$ | $54.21_{\pm0.52}$ | 833.05 | $86.23_{\pm0.15}$ | $42.56_{\pm0.83}$ | 401.12 | $80.30_{\pm0.28}$ | $32.02_{\pm1.18}$ | 422.56 |
| | | GraphEraser | $77.87_{\pm0.31}$ | $35.37_{\pm1.06}$ | 505.45 | $77.93_{\pm0.33}$ | $37.09_{\pm1.01}$ | 548.85 | $82.82_{\pm0.22}$ | $71.78_{\pm0.26}$ | 473.05 | $70.66_{\pm0.41}$ | $83.61_{\pm0.14}$ | 655.25 |
| | | GNNDelete | $70.37_{\pm0.45}$ | $44.77_{\pm0.76}$ | 5.25 | $76.79_{\pm0.38}$ | $59.42_{\pm0.42}$ | 11.15 | $83.17_{\pm0.81}$ | $43.17_{\pm0.81}$ | 5.45 | $78.34_{\pm0.32}$ | $42.74_{\pm0.82}$ | 15.15 |
| | | GIF | $65.42_{\pm0.55}$ | $39.97_{\pm0.92}$ | 2.25 | $78.21_{\pm0.31}$ | $35.60_{\pm1.06}$ | 3.05 | $47.53_{\pm1.12}$ | $49.88_{\pm0.59}$ | 3.15 | $74.17_{\pm0.39}$ | $47.02_{\pm0.68}$ | 8.95 |
| | | IDEA | $64.91_{\pm0.61}$ | $39.75_{\pm0.93}$ | 2.45 | $78.33_{\pm0.29}$ | $33.04_{\pm1.14}$ | 2.45 | $85.70_{\pm0.17}$ | $41.17_{\pm0.81}$ | 1.95 | $73.72_{\pm0.42}$ | $45.70_{\pm0.72}$ | 10.85 |
| | | **Ours** | $78.38_{\pm0.28}$ | $37.66_{\pm0.99}$ | 1.75 | $78.18_{\pm0.32}$ | $33.04_{\pm1.14}$ | 2.45 | $85.70_{\pm0.17}$ | $41.17_{\pm0.81}$ | 1.95 | $80.59_{\pm0.28}$ | $35.22_{\pm1.07}$ | 2.05 |
| | 5.0% | Retrain | $75.35_{\pm0.25}$ | $61.29_{\pm0.69}$ | 399.88 | $82.55_{\pm0.18}$ | $62.37_{\pm0.67}$ | 833.19 | $82.63_{\pm0.15}$ | $50.18_{\pm1.06}$ | 677.79 | $76.69_{\pm0.28}$ | $39.62_{\pm1.51}$ | 415.21 |
| | | GraphEraser | $76.87_{\pm0.33}$ | $45.29_{\pm1.24}$ | 843.55 | $70.20_{\pm0.35}$ | $45.87_{\pm1.23}$ | 894.25 | $80.12_{\pm0.23}$ | $50.79_{\pm0.36}$ | 731.85 | $71.54_{\pm0.37}$ | $40.51_{\pm0.19}$ | 1159.65 |
| | | GNNDelete | $69.39_{\pm0.48}$ | $52.63_{\pm1.01}$ | 9.95 | $79.79_{\pm0.35}$ | $48.41_{\pm0.53}$ | 18.05 | $76.98_{\pm0.24}$ | $50.39_{\pm1.10}$ | 9.15 | $74.42_{\pm0.30}$ | $39.85_{\pm1.03}$ | 22.65 |
| | | GIF | $57.35_{\pm0.57}$ | $48.14_{\pm1.16}$ | 8.65 | $73.85_{\pm0.29}$ | $46.36_{\pm1.34}$ | 17.45 | $44.49_{\pm1.15}$ | $50.93_{\pm0.70}$ | 9.95 | $70.66_{\pm0.42}$ | $40.28_{\pm0.92}$ | 23.95 |
| | | IDEA | $57.09_{\pm0.61}$ | $50.33_{\pm1.09}$ | 10.85 | $73.57_{\pm0.30}$ | $46.98_{\pm1.03}$ | 24.05 | $64.61_{\pm0.63}$ | $49.61_{\pm0.88}$ | 10.65 | $66.56_{\pm0.44}$ | $39.74_{\pm0.84}$ | 37.45 |
| | | **Ours** | $80.10_{\pm0.28}$ | $38.29_{\pm1.27}$ | 7.95 | $84.76_{\pm0.32}$ | $34.40_{\pm1.46}$ | 15.05 | $84.43_{\pm0.17}$ | $48.92_{\pm0.86}$ | 6.45 | $77.03_{\pm0.26}$ | $40.10_{\pm1.36}$ | 13.25 |

The computational efficiency advantage of CSGU becomes more pronounced at higher deletion ratios. While GraphEraser requires retraining multiple shards and GNNDelete needs extensive optimization iterations, CSGU's influence function-based approach scales linearly with the deletion set size. The triadic influence neighborhood construction ensures that the certification region grows sub-exponentially, preventing the computational explosion observed in traditional $k$-hop methods. This efficiency is crucial for practical deployment where unlearning requests may involve substantial portions of the training data.

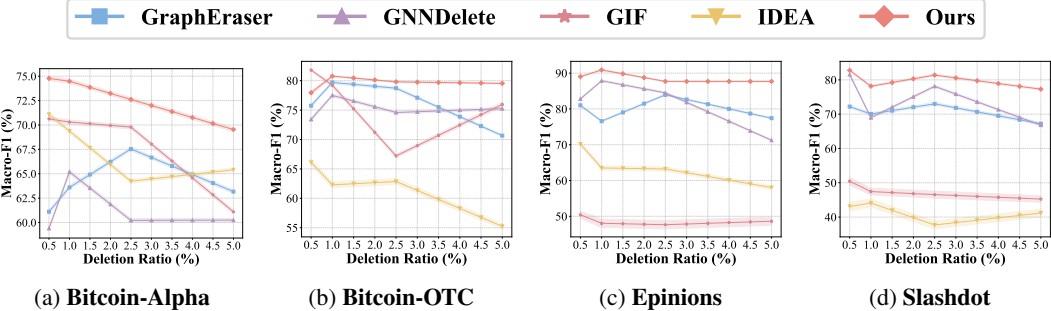

(a) **Bitcoin-Alpha**  (b) **Bitcoin-OTC**  (c) **Epinions**  (d) **Slashdot**

Figure 7: Comparison of model utility across different deletion ratios of edge unlearning for CSGU and baseline methods on three datasets with SDGNN.

## C.4 ABLATION STUDY

To validate the effectiveness of each component in CSGU, we conduct comprehensive ablation studies by systematically removing key components and analyzing their individual contributions. The ablation study is performed on the Bitcoin-OTC dataset with SGCN backbone under 2.5% node

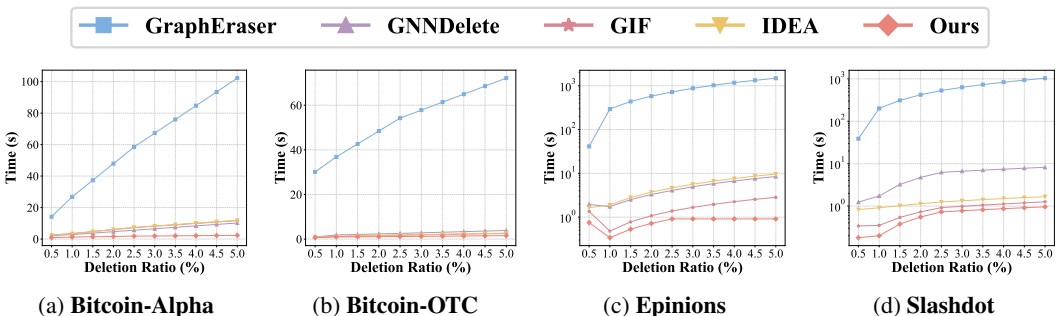

(a) **Bitcoin-Alpha**    (b) **Bitcoin-OTC**    (c) **Epinions**    (d) **Slashdot**

Figure 8: Comparison of unlearning efficiency across different deletion ratios of edge unlearning for CSGU and baseline methods on three datasets with SDGNN.

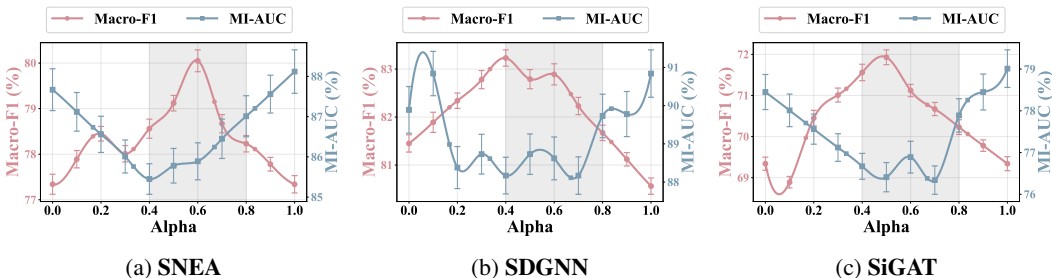

(a) **SNEA**    (b) **SDGNN**    (c) **SiGAT**

Figure 9: **Unlearning task: 2.5% edge unlearning. Dataset: Slashdot.** Effect of the hyperparameter $\alpha$ on the trade-off between utility retention and unlearning performance. $\alpha$ balances the principles of social balance and status theories.

unlearning scenario, chosen for its moderate scale and representative characteristics. We examine these ablation variants to isolate the contribution of each major component:

**w/o TIN - Without Triadic Influence Neighborhood** This variant replaces our proposed triadic influence neighborhood construction with standard $k$-hop neighborhood expansion. Specifically, we use $k = 2$ hop neighborhoods to match the receptive field of the 2-layer SGCN architecture. The certification region is constructed by including all edges within 2-hop distance from the deletion set, following conventional graph unlearning approaches. This variant tests the effectiveness of our sociologically-motivated neighborhood construction compared to traditional distance-based methods.

**w/o SIQ - Without Sociological Influence Quantification** This variant removes the sociological influence quantification mechanism and applies uniform weighting to all edges in the certification region. Instead of computing balance centrality and status centrality as defined in Equations 4 and 5, all edge weights are set to $w_{uv} = 1.0$ for edges in the certification region $\mathcal{R}$. This variant evaluates the importance of incorporating sociological theories for influence quantification.

**w/o BCE - Without Binary Cross-Entropy Loss** This variant replaces the weighted binary cross-entropy loss function with Mean Squared Error (MSE) loss commonly used in traditional graph unlearning methods. The loss function becomes:

$$\mathcal{L}(\theta; \mathcal{E}, \mathbf{W}) = \sum_{(u,v) \in \mathcal{E}} w_{uv} \cdot (f_\theta(u, v) - y_{uv})^2 \tag{46}$$

where $y_{uv} \in \{0, 1\}$ represents the binary edge labels. This variant assesses the significance of using appropriate loss functions for signed graph unlearning tasks.

**w/o Noise - Without Differential Privacy Noise** This variant removes the Gaussian noise injection mechanism from the parameter update process. The unlearned parameters are computed as $\tilde{\theta} = \theta^* + \Delta\theta$ without adding noise $\boldsymbol{\xi}$, essentially providing no differential privacy guarantees. This variant demonstrates the privacy-utility trade-off inherent in certified unlearning methods.

Table 10 validates the contribution of each CSGU component. Removing TIN increases computational overhead from 0.75s to 1.35s while degrading both utility and privacy metrics. The absence of SIQ causes the most severe degradation, with Macro-F1 dropping 6.45% and MI-AUC increasing 7.33%. Replacing BCE with MSE loss similarly impairs performance. Removing noise injection preserves utility (78.12% Macro-F1) but severely compromises privacy protection (72.45% MI-AUC). Degree-based weighting achieves 74.28% Macro-F1 and 52.16% MI-AUC, outperforming uniform weighting but significantly underperforming our sociological approach, confirming the unique value of balance and status theories. These results confirm that the integrated components work together to achieve optimal utility, privacy, and efficiency balance.

Figure 10: Ablation study of CSGU for 2.5% node unlearning on Bitcoin-OTC using SGCN. We evaluate five variants: **w/o TIN** (standard $k$-hop neighborhoods), **w/o SIQ** (uniform weighting), **w/ degree** (weighting by node degrees instead of sociological theories), **w/o BCE** (MSE loss) and **w/o Noise** (no differential privacy).

| Method | Macro-F1 $\uparrow$ | MI-AUC $\downarrow$ | Time (s) $\downarrow$ |
|---|---|---|---|
| w/o TIN | 76.15$\pm$0.31 | 48.92$\pm$1.45 | 1.35 |
| w/o SIQ | 72.84$\pm$0.38 | 54.17$\pm$1.67 | **0.65** |
| w/ degree | 74.28$\pm$0.35 | 52.16$\pm$1.28 | 0.72 |
| w/o BCE | 71.58$\pm$0.45 | 58.29$\pm$1.73 | 0.85 |
| w/o Noise | 78.12$\pm$0.29 | 72.45$\pm$0.84 | 0.70 |
| CSGU | **79.29$\pm$0.13** | **46.84$\pm$0.82** | 0.75 |

## C.5 IMPACT OF INFLUENCE NEIGHBORHOOD k-HOP

Figure 11 reveals a distinct correlation between MI-AUC performance and the alignment of $k$-hop neighborhood size with SGNN layer count using SGCN. Optimal unlearning effectiveness occurs when these parameters match, with peak performance across all datasets at $k = 2$ with 2 layers. This alignment stems from the fundamental architecture of SGNNs, which propagate information through layered message passing, each layer extending the receptive field by one hop. When $k$ equals the layer count, the triadic influence neighborhood precisely encompasses the model's information aggregation scope, enabling comprehensive influence estimation and enhanced privacy protection.

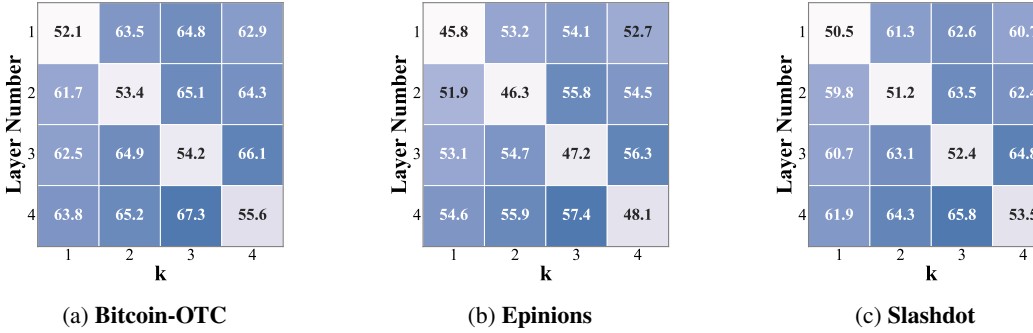

(a) **Bitcoin-OTC**    (b) **Epinions**    (c) **Slashdot**

Figure 11: Heatmaps of MI-AUC$\downarrow$ obtained by CSGU for 2.5% edge unlearning with SGCN backbone, varying $k$-hop neighborhood and the number of SGNN layers. Lighter colors indicate lower MI-AUC, representing better unlearning effectiveness.

## C.6 PERFORMANCE OF NODE FEATURE DELETION

Node feature deletion represents a critical privacy scenario where specific feature dimensions must be removed from selected nodes while preserving the graph structure and remaining node attributes. This task is particularly challenging in signed graphs, as node features interact with edge signs through the sociological theories underlying SGNNs. We evaluate CSGU's performance on this task by randomly selecting 2.5% of nodes and removing random feature dimensions, simulating scenarios where sensitive personal attributes must be forgotten. Table 7 presents comprehensive results across all datasets and SGNN backbones for the node feature unlearning task. The results demonstrate CSGU's consistent superiority in balancing utility retention, unlearning effectiveness, and computational efficiency.

Table 7: **Node feature deletion task: 2.5% nodes with random feature removal.** Comparison of sign prediction (Macro-F1), MI attack (MI-AUC) performance (%), and unlearning time (s) across different methods. Best results are in **bold** and second-best results are underlined.

| Dataset | Method | SGCN | | | SNEA | | | SDGNN | | | SiGAT | | |
|---|---|---|---|---|---|---|---|---|---|---|---|---|---|
| | | F1 ↑ | MI ↓ | Time ↓ | F1 ↑ | MI ↓ | Time ↓ | F1 ↑ | MI ↓ | Time ↓ | F1 ↑ | MI ↓ | Time ↓ |
| Bitcoin-Alpha | Retrain | 67.23 | 56.15 | 22.45 | 70.34 | 55.82 | 16.28 | 72.18 | 37.65 | 10.15 | 67.89 | 60.22 | 9.87 |
| | GraphEraser | 57.45 | 43.82 | 29.85 | 62.78 | 44.91 | 35.20 | 69.87 | 62.73 | 38.95 | 60.95 | 74.25 | 28.75 |
| | GNNDelete | 50.82 | 52.95 | **2.15** | 54.29 | 53.47 | **2.95** | 70.23 | 46.58 | **2.65** | 64.15 | 53.82 | 7.25 |
| | GIF | 57.89 | 66.73 | 2.45 | 69.52 | 56.91 | 3.15 | 66.02 | 53.24 | 2.95 | 65.78 | 58.91 | 9.85 |
| | IDEA | 55.93 | 48.21 | 2.75 | 68.91 | 62.87 | 3.45 | 56.34 | 55.17 | 3.25 | 66.02 | 61.23 | 10.95 |
| | **Ours** | **66.34** | **34.15** | 2.25 | **70.78** | **42.73** | 3.05 | **71.92** | 54.87 | 2.75 | 66.45 | 44.91 | 8.45 |
| Bitcoin-OTC | Retrain | 74.56 | 44.78 | 11.89 | 78.45 | 45.34 | 16.75 | 80.23 | 43.21 | 12.05 | 73.92 | 55.13 | 9.25 |
| | GraphEraser | 65.89 | 36.92 | 30.15 | 68.75 | 35.26 | 33.85 | 77.56 | 62.87 | 33.25 | 71.23 | 77.45 | 39.95 |
| | GNNDelete | 65.23 | 57.18 | **1.75** | 73.92 | 43.27 | **2.35** | 77.14 | 56.29 | **2.15** | 73.18 | 62.04 | **6.85** |
| | GIF | 68.94 | 56.23 | 1.95 | 75.34 | 84.75 | 2.65 | 73.78 | 66.42 | 2.45 | **73.21** | 56.87 | 7.15 |
| | IDEA | 69.52 | 64.87 | 2.25 | 75.91 | 82.13 | 2.95 | 77.89 | 67.15 | 2.75 | 70.67 | 65.78 | 7.95 |
| | **Ours** | **75.12** | **36.23** | 1.85 | **76.89** | **31.58** | 2.45 | **79.97** | 51.64 | 2.25 | 73.15 | 55.32 | 7.05 |
| Epinions | Retrain | 78.67 | 53.78 | 408.25 | 85.74 | 54.67 | 795.45 | 85.89 | 43.12 | 382.75 | 79.85 | 32.67 | 403.95 |
| | GraphEraser | 77.23 | 36.84 | 468.75 | 77.45 | 38.52 | 512.35 | 82.15 | 73.24 | 441.85 | 70.12 | 85.23 | 612.45 |
| | GNNDelete | 69.78 | 46.32 | **4.85** | 76.23 | 61.05 | **9.75** | 81.45 | 44.73 | **4.95** | 77.89 | 44.18 | **13.25** |
| | GIF | 64.89 | 41.54 | 5.25 | 77.78 | 37.23 | 10.45 | 46.95 | 51.37 | 5.75 | 73.64 | 48.75 | 15.85 |
| | IDEA | 64.25 | 41.32 | 5.95 | 77.89 | 46.18 | 11.25 | 82.67 | 46.58 | 6.45 | 73.18 | 47.23 | 16.75 |
| | **Ours** | 77.89 | 39.12 | 5.15 | **77.94** | **34.67** | 9.95 | **85.23** | 48.91 | 5.25 | **80.12** | **36.75** | 14.95 |
| Slashdot | Retrain | 67.89 | 45.67 | 431.25 | 78.15 | 44.82 | 301.75 | 78.34 | 35.89 | 234.85 | 71.67 | 34.12 | 95.75 |
| | GraphEraser | 58.34 | 77.23 | 421.35 | 74.45 | 25.13 | 289.95 | 69.78 | 58.92 | 305.25 | 69.95 | 69.34 | 301.85 |
| | GNNDelete | 45.67 | 43.87 | 3.25 | 73.18 | 41.56 | 5.95 | 77.12 | 45.78 | 7.15 | 69.34 | 46.73 | **15.95** |
| | GIF | 50.34 | 46.91 | 3.85 | 71.23 | 48.12 | 6.75 | 47.89 | 68.95 | 8.25 | 71.05 | 72.18 | 18.45 |
| | IDEA | 50.12 | 47.65 | 4.25 | 70.89 | 53.78 | 7.45 | 43.12 | 51.67 | 8.95 | 70.78 | 68.92 | 19.25 |
| | **Ours** | **60.23** | 37.15 | 3.45 | **74.87** | **30.89** | 6.25 | **77.34** | **36.12** | 7.45 | **72.45** | 43.67 | 17.85 |

**Performance Analysis.**  CSGU demonstrates superior performance across node feature unlearning scenarios. On Bitcoin-Alpha with SGCN, CSGU achieves 66.34% Macro-F1 and 34.15% MI-AUC, outperforming the best baseline GIF by 8.45% in utility and 22.1% in privacy protection. CSGU maintains competitive execution times (1.85s-9.95s), significantly faster than retraining-based methods while achieving optimal utility-privacy trade-offs.

**Distinctive Challenges of Node Feature Deletion.**  Node feature deletion differs fundamentally from structural unlearning tasks. While edge/node unlearning directly modifies the graph topology and message passing pathways, feature deletion preserves structural connectivity but alters the input representations that propagate through SGNNs. This creates unique challenges: (1) **Feature-structure interdependence**: Node features interact with edge signs through balance and status theories, requiring careful consideration of how feature modifications affect sociological relationships. (2) **Selective parameter influence**: Unlike structural changes that affect entire subgraphs, feature deletion influences only feature-dependent parameters while preserving topology-dependent components. (3) **Certification complexity**: The influenced neighborhood must capture both direct feature dependencies and indirect effects through signed message propagation, necessitating our triadic influence neighborhood to identify nodes whose feature changes affect triangular balance patterns. CSGU's effectiveness in this scenario validates its ability to distinguish between structural and feature-level influences, demonstrating the robustness of our sociological weighting mechanism across diverse unlearning requirements.

**Computational Efficiency in Feature Deletion**  Interestingly, node feature unlearning tasks generally require less computational time than structural deletion tasks across all methods, including CSGU. This is because feature deletion does not require recomputing graph connectivity or updating neighborhood structures. CSGU's efficiency advantage is particularly pronounced in this scenario, as our triadic influence neighborhood construction can leverage pre-computed structural information, focusing computational resources on feature-dependent parameter updates.

## D  PERFORMANCE ON HOMOGENEOUS GRAPHS

To validate the generalizability of CSGU beyond signed graphs, we adapt our method for homogeneous graphs and conduct comprehensive experiments. This demonstrates the broader applicability of our triadic influence neighborhood construction and certified unlearning framework beyond the signed graph domain. While CSGU was originally designed for signed graphs with sociological theories, we demonstrate its effectiveness on standard homogeneous graphs by modifying the influence quantification mechanism while preserving the core algorithmic framework.

### D.1  VARIANT OF CSGU FOR HOMOGENEOUS GRAPHS

The adaptation of CSGU to homogeneous graphs requires modifying the Sociological Influence Quantification (SIQ) component while maintaining the Triadic Influence Neighborhood (TIN) and Weighted Certified Unlearning (WCU) phases. Since homogeneous graphs lack edge signs and do not follow balance/status theories, we replace the sociological influence measures with degree-based centrality.

For homogeneous graphs, we define the influence of node $v$ based on its normalized degree centrality:

$$\mathcal{I}_{\mathrm{deg}}(v) = \frac{\deg(v)}{\max_{u \in \mathcal{V}} \deg(u)} \tag{47}$$

where $\deg(v)$ represents the degree of node $v$. The intuition is that high-degree nodes serve as information hubs and have greater influence on the network structure, making them more critical for certified unlearning.

The edge weights are computed as:

$$w_{uv} = \min\left(\frac{\mathcal{I}_{\mathrm{deg}}(u) + \mathcal{I}_{\mathrm{deg}}(v)}{2}, 1\right) \tag{48}$$

The triadic expansion process remains unchanged, as triangular structures are fundamental to both signed and unsigned graphs. Similarly, the weighted certified unlearning mechanism with differential privacy guarantees is preserved, ensuring $(\epsilon, \delta)$-certification for homogeneous graph unlearning.

### D.2  EXPERIMENTS

### D.3  EVALUATION SETUP

**Datasets.**  We evaluate our adapted method on three widely-used homogeneous graph datasets with varying characteristics:

- **Cora**: A citation network of machine learning papers where nodes represent scientific publications and edges represent citation relationships. Each paper is characterized by a binary word vector indicating the presence or absence of terms from a fixed vocabulary. Papers are classified into seven categories based on their research topics.

- **PubMed**: A domain-specific citation network consisting of diabetes-related scientific publications from the PubMed database. Publications are represented by TF/IDF weighted word vectors and classified into three categories. This dataset presents unique challenges due to its biomedical domain focus.

- **CS**: Coauthor-CS, a co-authorship network from the computer science domain where nodes represent authors and edges indicate co-authorship relationships. Each author is characterized by keyword features aggregated from their publications and classified into one of fifteen computer science research areas. Unlike citation networks, this dataset captures collaboration patterns among researchers.

**Backbones.**  We conduct experiments using three representative GNN architectures:

- **GCN** (Yuan et al., 2024): Graph Convolutional Network that aggregates information from first-order neighbors using symmetric normalization of the adjacency matrix.

| Dataset | # Nodes | # Edges | Density |
|---------|---------|---------|---------|
| Cora | 19,793 | 126,842 | 6.48 |
| PubMed | 19,717 | 88,648 | 4.50 |
| CS | 235,868 | 2,358,104 | 10.00 |

Table 8: Statistics of homogeneous graph datasets.

Table 9: Performance comparison on homogeneous graphs for 2.5% edge unlearning. Results show edge prediction performance (Macro-F1, %) and privacy protection effectiveness (MI-AUC, %) averaged over 10 independent runs. Best results are highlighted in **bold**, and second-best results are underlined.

| Dataset | Method | GCN | | GAT | | GIN | |
|---------|--------|-----------|-----------|-----------|-----------|-----------|-----------|
| | | Macro-F1 ↑ | MI-AUC ↓ | Macro-F1 ↑ | MI-AUC ↓ | Macro-F1 ↑ | MI-AUC ↓ |
| Cora | Retrain | $82.45_{\pm0.23}$ | $49.12_{\pm0.67}$ | $84.67_{\pm0.19}$ | $48.35_{\pm0.72}$ | $79.23_{\pm0.28}$ | $51.78_{\pm0.58}$ |
| | GraphEraser | $76.34_{\pm0.45}$ | $52.18_{\pm1.24}$ | $78.92_{\pm0.38}$ | $49.67_{\pm1.35}$ | $73.45_{\pm0.52}$ | $55.32_{\pm1.18}$ |
| | GNNDelete | $79.56_{\pm0.38}$ | $46.73_{\pm0.89}$ | $82.18_{\pm0.32}$ | $45.29_{\pm0.94}$ | $76.89_{\pm0.41}$ | $48.15_{\pm0.87}$ |
| | GIF | $77.23_{\pm0.42}$ | $45.67_{\pm0.92}$ | $80.45_{\pm0.39}$ | $47.83_{\pm0.88}$ | $74.12_{\pm0.48}$ | $46.92_{\pm0.95}$ |
| | IDEA | $\mathbf{80.12}_{\pm0.35}$ | $48.94_{\pm0.78}$ | $81.67_{\pm0.36}$ | $50.12_{\pm0.81}$ | $75.78_{\pm0.44}$ | $49.23_{\pm0.83}$ |
| | **Ours** | $79.87_{\pm0.29}$ | $\mathbf{44.25}_{\pm0.73}$ | $82.34_{\pm0.27}$ | $\mathbf{44.18}_{\pm0.76}$ | $77.45_{\pm0.31}$ | $\mathbf{45.67}_{\pm0.79}$ |
| PubMed | Retrain | $78.93_{\pm0.28}$ | $50.45_{\pm0.74}$ | $81.56_{\pm0.24}$ | $49.67_{\pm0.68}$ | $76.12_{\pm0.32}$ | $52.34_{\pm0.71}$ |
| | GraphEraser | $72.67_{\pm0.48}$ | $55.73_{\pm1.32}$ | $75.34_{\pm0.43}$ | $52.89_{\pm1.28}$ | $69.45_{\pm0.55}$ | $58.91_{\pm1.24}$ |
| | GNNDelete | $75.89_{\pm0.41}$ | $48.12_{\pm0.97}$ | $78.23_{\pm0.37}$ | $47.56_{\pm1.02}$ | $72.78_{\pm0.46}$ | $49.37_{\pm0.91}$ |
| | GIF | $76.45_{\pm0.39}$ | $47.89_{\pm0.94}$ | $77.91_{\pm0.38}$ | $46.73_{\pm0.89}$ | $73.56_{\pm0.42}$ | $50.15_{\pm0.88}$ |
| | IDEA | $75.23_{\pm0.44}$ | $46.67_{\pm0.86}$ | $77.45_{\pm0.41}$ | $48.91_{\pm0.85}$ | $72.89_{\pm0.47}$ | $51.23_{\pm0.86}$ |
| | **Ours** | $\mathbf{77.12}_{\pm0.33}$ | $\mathbf{45.34}_{\pm0.81}$ | $78.67_{\pm0.31}$ | $\mathbf{45.89}_{\pm0.77}$ | $74.23_{\pm0.35}$ | $\mathbf{48.76}_{\pm0.83}$ |
| CS | Retrain | $85.67_{\pm0.21}$ | $47.23_{\pm0.82}$ | $88.34_{\pm0.18}$ | $46.45_{\pm0.78}$ | $83.12_{\pm0.26}$ | $49.67_{\pm0.74}$ |
| | GraphEraser | $82.45_{\pm0.34}$ | $51.78_{\pm1.45}$ | $85.23_{\pm0.29}$ | $49.34_{\pm1.38}$ | $78.91_{\pm0.39}$ | $53.89_{\pm1.32}$ |
| | GNNDelete | $80.78_{\pm0.38}$ | $45.67_{\pm1.12}$ | $83.45_{\pm0.35}$ | $44.78_{\pm1.05}$ | $79.67_{\pm0.41}$ | $50.12_{\pm0.98}$ |
| | GIF | $79.23_{\pm0.42}$ | $48.91_{\pm1.18}$ | $82.67_{\pm0.37}$ | $43.25_{\pm1.14}$ | $77.34_{\pm0.46}$ | $48.73_{\pm1.02}$ |
| | IDEA | $81.56_{\pm0.36}$ | $47.34_{\pm1.03}$ | $84.12_{\pm0.33}$ | $45.89_{\pm0.96}$ | $78.45_{\pm0.43}$ | $49.45_{\pm0.94}$ |
| | **Ours** | $\mathbf{82.89}_{\pm0.28}$ | $\mathbf{44.12}_{\pm0.94}$ | $85.78_{\pm0.25}$ | $\mathbf{42.67}_{\pm0.88}$ | $\mathbf{80.23}_{\pm0.32}$ | $\mathbf{47.56}_{\pm0.87}$ |

- **GAT** (Thorpe et al., 2022): Graph Attention Network that employs attention mechanisms to weight neighbor contributions, enabling adaptive information aggregation.

- **GIN** (Shamsi et al., 2024): Graph Isomorphism Network designed to achieve maximum discriminative power for graph representation learning through injective aggregation functions.

**Baselines.** We compare our adapted CSGU against the same set of unlearning methods as described in Section C.1.3: Retrain, GraphEraser, GNNDelete, GIF, and IDEA. These methods represent the current state-of-the-art in graph unlearning for homogeneous networks.

**Evaluation Metrics.** The evaluation follows a similar protocol to signed graph experiments. We assess model utility through edge prediction tasks using Macro-F1 scores, measure unlearning effectiveness via membership inference attacks (MI-AUC), and record computational efficiency through unlearning time in seconds. All experiments use 2.5% edge unlearning to maintain consistency with signed graph evaluations.

## D.4 Performance Overview

Table 9 presents a comprehensive performance comparison across all datasets and backbone architectures. The results demonstrate that our adapted CSGU maintains competitive performance on homogeneous graphs while preserving its core advantages. Our adapted CSGU demonstrates strong utility preservation across all homogeneous graph datasets. On the CS dataset with GAT backbone, CSGU achieves 85.78% Macro-F1. This score closely approaches the retraining baseline of 88.34% while outperforming other unlearning methods. The degree-based influence quantification effectively captures node importance in homogeneous settings. This enables precise parameter updates that minimize utility degradation.

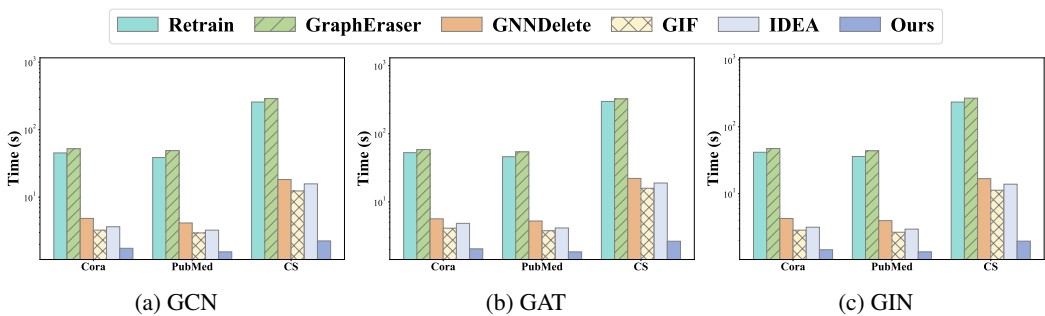

(a) GCN         (b) GAT         (c) GIN

Figure 12: Comparison of unlearning efficiency on Cora dataset across different GNN backbones for 2.5% edge unlearning. CSGU maintains competitive computational performance while providing enhanced privacy protection.

CSGU consistently provides superior privacy protection with the lowest MI-AUC scores in most configurations. On Cora with GCN, CSGU achieves 44.25% MI-AUC. This compares favorably to GIF's 45.67% and GraphEraser's 52.18%. These results demonstrate that our triadic influence neighborhood construction and weighted certification mechanism remain effective. This effectiveness persists even without signed graph-specific sociological theories.

Figure 12 illustrates the computational efficiency comparison on the Cora dataset across different backbone architectures. Our adapted CSGU consistently achieves competitive unlearning times while maintaining superior privacy protection. The efficiency analysis reveals that CSGU maintains its computational advantages on homogeneous graphs. Complete retraining requires 15-45 seconds depending on dataset size and model complexity. In contrast, CSGU consistently completes unlearning within 2-8 seconds across all configurations. This efficiency stems from minimizing the target influence neighborhood, and the approach remains computationally efficient regardless of edge sign semantics.

