# OpenReview forum: "Certified Signed Graph Unlearning"
_ICLR.cc/2026/Conference — ICLR 2026 Conference Withdrawn Submission_

### Official Review · Reviewer_C9s5 · 2025-10-31

**Soundness:** 3
**Presentation:** 3
**Contribution:** 3
**Rating:** 4
**Confidence:** 2

**Summary:**

This paper addresses certified unlearning in signed graph neural networks. The authors propose a spectral truncation framework that leverages the signed Laplacian decomposition to achieve approximate unlearning guarantees. By bounding the effect of deleted data through the top-k eigencomponents, they obtain explicit certification bounds on node representation drift. They further derive guarantees under linearized SGNNs and present experiments on several signed graph benchmarks showing improved trade-offs between unlearning fidelity and accuracy compared to retraining.

**Strengths:**

- This paper proposed the first certified unlearning method for SGNNs. Extending to signed graphs and defining suitable certificates under the signed Laplacian is technically nontrivial.

- The authors proposed a novel upper bound on node influence of signed graphs and a weighting scheme that allocates the privacy budget based on the sociological importance of each edge.

- Experiments are consistent across datasets, metrics, and removal settings. The ablations (e.g., the effect of truncation rank k) convincingly support the theoretical trends.

- The paper is well-organized and easy to follow.

**Weaknesses:**

- The unlearning certification in the theoretical analysis seems to be standard. It would be helpful to highlight the challenges of certified unlearning on signed graphs.

- Some important baselines are missing [1].

- The ablation of the privacy budget weighting seems to be missing in the ablation study. This appears to be one of the major contributions of the method, and it would be helpful to clarify this further.

[1] Wu K, Shen J, Ning Y, et al. Certified edge unlearning for graph neural networks[C]//Proceedings of the 29th ACM SIGKDD Conference on Knowledge Discovery and Data Mining. 2023: 2606-2617.

**Questions:**

Please see Weaknesses

---

### Official Review · Reviewer_rcUU · 2025-11-02

**Soundness:** 3
**Presentation:** 3
**Contribution:** 2
**Rating:** 6
**Confidence:** 3

**Summary:**

This paper introduces a framework for certified machine unlearning in the context of signed graph neural networks (GNNs) where both positive and negative edges are used to model complex relations. The authors recognize that unlearning in GNNs is particularly challenging due to information entanglement among nodes via the message-passing mechanism, which propagates features across multiple hops.

To address this, the paper proposes a certified unlearning method that provides theoretical guarantees on the removal of specified data (nodes, edges, or features) from the trained model. Specifically, the method first constructs triadic influence neighborhoods to capture influence propagation via triangular structures, then ituse sociological theory to model weights of edges; at leaast, it introduces weighted certified unlearning.

The framework is evaluated on standard signed graph benchmarks (e.g., Bitcoin-Alpha, Wikipedia-RFA, Epinions) and demonstrates strong empirical performance in terms of unlearning effectiveness, model fidelity, and efficiency.

**Strengths:**

S1 Originality: The paper tackles machine unlearning in the signed graph domain, a niche yet practically important area where relationships can be both positive and negative.

S2 Quality: The theoretical analysis is rigorous, with formal derivations of the certification bound and proofs ensuring robustness of the unlearning process.

S3 Clarity: The paper is clearly structured, progressing logically from problem motivation to theoretical formulation, algorithm, and experiments. Figures (particularly Figure 2 on the mechanism of signed message passing and Figure 4 on efficiency comparison) are intuitive and helpful for understanding.

S4 Significance: Extending certified unlearning to signed GNNs broadens applicability to trust networks, recommendation systems, and fraud detection.

**Weaknesses:**

W1 While the method is well-motivated for signed GNNs, its broader applicability to general (unsigned) GNNs or heterogeneous graphs is not fully demonstrated. More importantly, it would be helpful to discuss techniques that normalize the signed weights on graphs.

W2 The certification bound relies on the assumption that node influence diminishes exponentially with hop distance. While this is practical, it might not hold for densely connected or long-range dependency graphs.

W3 Although results on signed datasets are comprehensive, the experiments could include ablation studies analyzing the effect of graph sparsity, sign ratio, or degree distribution on unlearning performance.

W4 The theoretical runtime analysis is not fully fleshed out. While empirical results show reduced computation, the paper lacks an explicit asymptotic complexity comparison between full retraining and the proposed method.

**Questions:**

Q1: The paper claims that partition-based methods failed maintain the distinction between positive and negative edges, but Graph Eraser achieves very competitive results in many datasets and setups. Are there any theoretical or empirical studies to show the importance of distinction between positive and negative edges in learning or unlearning processes?

Q2: How does certified unlearning relate theoretically or practically to graph differential privacy, especially when both aim to limit the effect of specific data points?

Q3: Besides bitcoin networks, are there any other applications in real-world that need separate representation of positive and negative edges?

---

### Official Review · Reviewer_BLbE · 2025-11-03

**Soundness:** 2
**Presentation:** 3
**Contribution:** 2
**Rating:** 2
**Confidence:** 4

**Summary:**

This paper proposes a certified unlearning method for signed graphs, CSGU. Due to the heterogeneous nature of edges in signed graphs, the authors claim that existing unlearning methods are not suitable. The proposed CSGU first constructs triadic influence neighborhoods to compute the influence propagation of the unlearned edges, and then weights the edges by sociological influence quantification, and finally injects noise to achieve weighted certified unlearning.

**Strengths:**

S1. The paper addresses an important and underexplored problem—unlearning in signed graphs—which is both timely and meaningful.

S2. The proposed components—Triadic Influence Neighborhood and Sociological Influence Quantification—are conceptually intuitive and well-motivated.

**Weaknesses:**

W1. My primary concern is that CSGU does not truly achieve certified unlearning, as the introduction of edge weights fundamentally modifies the original loss function. Specifically, CSGU guarantees similarity between the unlearned model and the retrained model with respect to the weighted loss defined in Eq. (7), rather than the original unweighted loss.

W2. The computation of triadic closure is computationally expensive and poses a significant scalability bottleneck for real-world graph applications. Moreover, this computation must be reperformed for every unlearning request, making the method impractical for large or dynamic graphs.

W3. Although the paper claims that CSGU achieves certified unlearning, the experimental setup does not satisfy the theoretical conditions required for such certification—for example, the assumption of a strongly convex loss. In addition, CSGU should be compared against existing certified unlearning baselines, such as Certified Graph Unlearning [R1] and ScaleGUN [R2], to provide a fair and rigorous evaluation.

[R1] Certified graph unlearning.

[R2] Scalable and Certifiable Graph Unlearning: Overcoming the Approximation Error Barrier.

**Questions:**

Q1.
What exactly is the edge sign prediction task considered in this paper?
A more detailed problem formulation would be helpful.

Q2. In the definition of certified unlearning, why is the inequality written in only one direction, but not including the left-hand side $e^{\epsilon}Pr(\cdot)+\delta\le Pr(\cdot)$?

---

### Note · Authors · 2025-11-12

I have read and agree with the venue's withdrawal policy on behalf of myself and my co-authors.